# Epithelial-to-mesenchymal transition proceeds through directional destabilization of multidimensional attractor

**Weikang Wang[1]\*, Dante Poe[1,2], Yaxuan Yang[1], Thomas Hyatt[3], Jianhua Xing[1,3,4]\***

[1]Department of Computational and Systems Biology, University of Pittsburgh, Pittsburgh, United States; [2]Joint CMU-Pitt Ph.D. Program in Computational Biology, University of Pittsburgh, Pittsburgh, United States; [3]Department of Physics and Astronomy, University of Pittsburgh, Pittsburgh, United States; [4]UPMC-Hillman Cancer Center, University of Pittsburgh, Pittsburgh, United States

**\*For correspondence:**
weikang@pitt.edu (WW);
xing1@pitt.edu (JX)

**Competing interest:** The authors declare that no competing interests exist.

**Abstract** How a cell changes from one stable phenotype to another one is a fundamental problem in developmental and cell biology. Mathematically, a stable phenotype corresponds to a stable attractor in a generally multi-dimensional state space, which needs to be destabilized so the cell relaxes to a new attractor. Two basic mechanisms for destabilizing a stable fixed point, pitchfork and saddle-node bifurcations, have been extensively studied theoretically; however, direct experimental investigation at the single-cell level remains scarce. Here, we performed live cell imaging studies and analyses in the framework of dynamical systems theories on epithelial-to-mesenchymal transition (EMT). While some mechanistic details remain controversial, EMT is a cell phenotypic transition (CPT) process central to development and pathology. Through time-lapse imaging we recorded single cell trajectories of human A549/Vim-RFP cells undergoing EMT induced by different concentrations of exogenous TGF-β in a multi-dimensional cell feature space. The trajectories clustered into two distinct groups, indicating that the transition dynamics proceeds through parallel paths. We then reconstructed the reaction coordinates and the corresponding quasi-potentials from the trajectories. The potentials revealed a plausible mechanism for the emergence of the two paths where the original stable epithelial attractor collides with two saddle points sequentially with increased TGF-β concentration, and relaxes to a new one. Functionally, the directional saddle-node bifurcation ensures a CPT proceeds towards a specific cell type, as a mechanistic realization of the canalization idea proposed by Waddington.

## Editor's evaluation

This is a multifaceted study of the epithelial to mesenchymal transition (EMT) in live cells. EMT is relevant for cancer, development, and wound healing. The authors were able to discern two possible cell transition path categories without multi-color labeling or other advanced experimental approaches, which could be impactful for other studies. The study draws on a wide range of experimental, data science, and modelling tools and techniques.

## Introduction

Cells of multicellular organisms assume different phenotypes that can have drastically different functions, morphologies, and gene expression patterns, and can undergo distinct changes when

**eLife digest** Cells with the same genetic code can take on many different formss, or phenotypes, which have distinct roles and appearances. Sometimes cells switch from one phenotype to another as part of healthy growth or during disease. One such change is the epithelial-to-mesenchymal transition (EMT), which is involved in fetal development, wound healing and the spread of cancer cells.

During EMT, closely connected epithelial cells detach from one another and change into mesenchymal cells that are able to migrate. Cells undergo a number of changes during this transition; however, the path they take to reach their new form is not entirely clear. For instance, do all cells follow the same route, or are there multiple ways that cells can shift from one state to the next?

To address this question, Wang et al. studied individual lung cancer cells that had been treated with a protein that drives EMT. The cells were then imaged at regular intervals over the course of two to three days to see how they changed in response to different concentrations of protein. Using a mathematical analysis designed to study chemical reactions, Wang et al. showed that the cells transform into the mesenchymal phenotype through two main routes.

This result suggests that attempts to prevent EMT, in cancer treatment for instance, would require blocking both paths taken by the cells. This information could be useful for biomedical researchers trying to regulate the EMT process. The quantitative approach of this study could also help physicists and mathematicians study other types of transition that occur in biology.

subjected to specific stimuli and microenvironments. Examples of cell phenotypic transitions (CPTs) include cell differentiation during development and induced and spontaneous cell fate transition such as reprogramming and trans-differentiation. Epithelial-to-mesenchymal transition (EMT) is a prototypic progress that has been extensively studied due to its significance in cell and developmental biology as well as in biomedical research (*Figure 1a*). Recent advances in single-cell techniques have further accelerated the long-term efforts on unraveling the mechanisms of CPTs, for understanding processes such as differentiation and reprogramming in developmental and cell biology, and for potential biomedical implications of modulating cell phenotypes in regenerative medicine and diseases such as cancer and chronic diseases (*Wagner and Klein, 2020*; *Weinreb et al., 2020*).

Considering cell as a dynamical system, understanding the CPT process from a dynamical systems theory perspective is an intriguing long-standing question in mathematical and systems biology. Mathematically, at any time a cell state can be specified by a set of state variables, such as gene expression levels, or other collective cell feature variables. A stable cell phenotype is an attractor in a high-dimensional cell state space formed by the state variables (*Huang et al., 2005*), and a CPT is a transition between different attractors. Waddington famously made an analogy comparing developmental processes to a ball sliding down a potential landscape, and diverging into multiple paths at some bifurcation points (*Rand et al., 2021*). This picture exemplifies a pitchfork bifurcation, where an original stable attractor turns to an unstable saddle point together with the emergence of two new attractors (*Figure 1b* top). Another well-discussed mechanism for attractor-to-attractor transition is through a series of saddle-node bifurcations, where first a new fixed point emerges, splits into a saddle point and attractor, then the saddle point separating the two attractors moves toward and eventually collides with the old attractor to destabilize the latter (*Figure 1b* bottom). A notable difference between the two types of bifurcations is that after a pitchfork bifurcation the original state can relax to multiple new stable states, while the saddle-node bifurcation the system is directed to relax to one attractor. A fundamental theoretical question is which mechanism a CPT process assumes, and why.

Pitchfork and saddle-node bifurcation are two important theoretical mechanisms of critical state transition and have been discussed in the context of CPTs (*Mojtahedi et al., 2016*). A new challenge then is how to evaluate various mathematical models experimentally at single cell levels. Several studies have suggested that analyzing snapshot single-cell genomics data (*Mojtahedi et al., 2016*; *Chen et al., 2012*), fixed cell data, to understand critical state transitions cannot provide temporal information on how individual cells transition. In addition, information from fluorescence-based live cell imaging is typically restricted to a small number of molecular species. Thus, tracking a small number

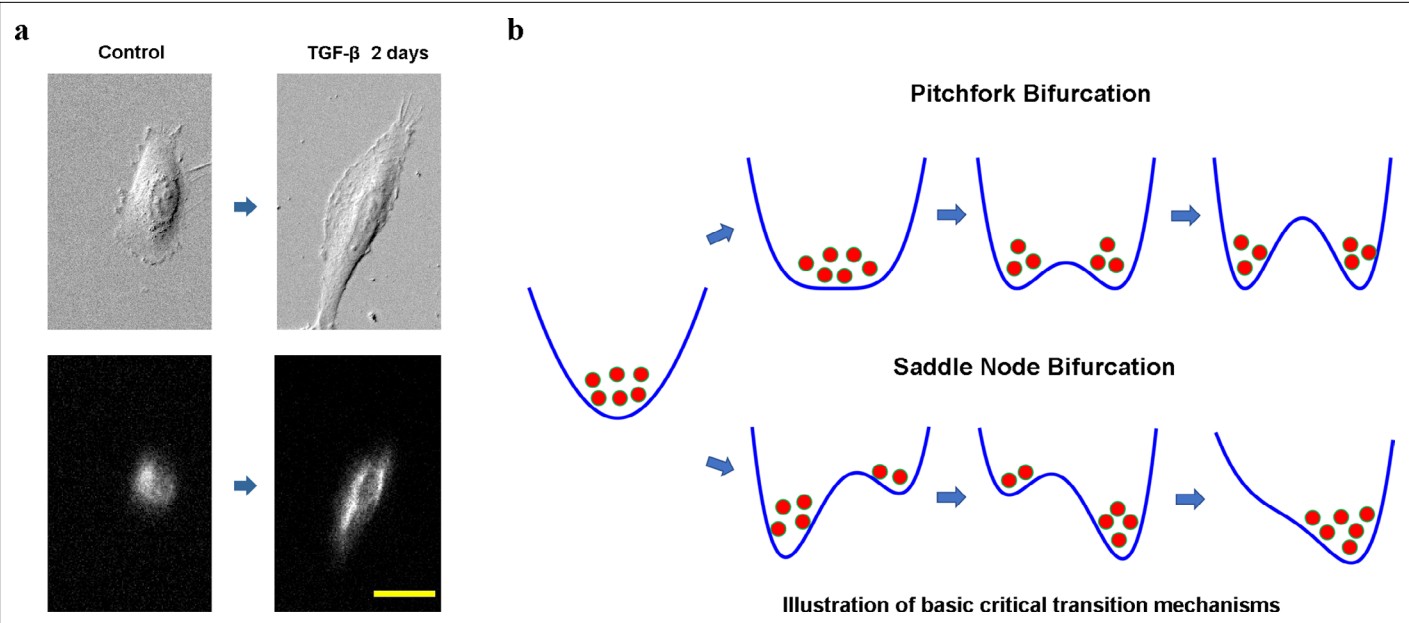

**Figure 1.** Cell phenotypic transitions as critical state transitions. (**a**) Transmission light and fluorescence (Vimentin-RFP) images showing an A549/Vim-RFP cell undergoing epithelial-to-mesenchymal transition. Scale bar: 30 µm. (**b**) Possible critical transition mechanisms in EMT. This is an illustration of pitchfork (top) and saddle-node (bottom) bifurcations using 1-D potential systems.

of molecular species through live cell imaging cannot provide the collective transition dynamics due to the intrinsic high-dimensional nature.

Specifically in the context of EMT, theoretical studies suggest that EMT proceeds as a saddle-node bifurcation (*Tian et al., 2013*); however, there is no direct experimental study on the single-cell transition dynamics. As recognized in a consensus statement from researchers in the EMT field, several open questions and challenges exist on understanding the mechanisms of the EMT process (*Yang et al., 2020*). For example, it is unclear whether the process proceeds as hopping among a small number of discrete and distinct intermediate states, or a continuum of such states with no clear boundary. The transition may proceed either along a linear array of states or through multiple parallel paths. Pseudo-time analyses of high throughput single cell genomics studies infer that EMT proceeds through a 1-D continuum path (*McFaline-Figueroa et al., 2019*), consistent with a prevalent EMT axis concept with the epithelial and the mesenchymal states as the two end states (*Nieto et al., 2016*). These predictions, which are indirectly inferred from snapshot single-cell data, require direct testing by tracking single cells over time. However, live cell imaging studies are impeded because EMT status cannot be assessed based on only a small number of molecular markers such as key transcriptional factors (*Yang et al., 2020*).

To address the above challenge when studying EMT dynamics, recently we developed a platform of tracking cell state change in a composite cell feature space that is accessible for multiplex and long-term live cell imaging (*Wang et al., 2020*). Here, we first apply the platform to study EMT in a human A549 derivative cell line with endogenous vimentin-RFP labeling (ATCC CCL-185EMT, denoted as A549/Vim-RFP in later discussions) induced by different concentrations of TGF-β. Then we analyze an ensemble of recorded multi-dimensional single-cell trajectories within the framework of reaction rate theories that have been a focused subject in the context of physics and chemistry.

## Results

### Single-cell trajectories are mathematically represented in a collective morphology/texture feature space

For quantitative studies of CPT dynamics, one first needs to establish a mathematical representation of the cell states and cell trajectories. Mathematically, one can represent a cell state by a point in a multi-dimensional space defined by gene expression (*Ye and Sarkar, 2018*) or other cell properties

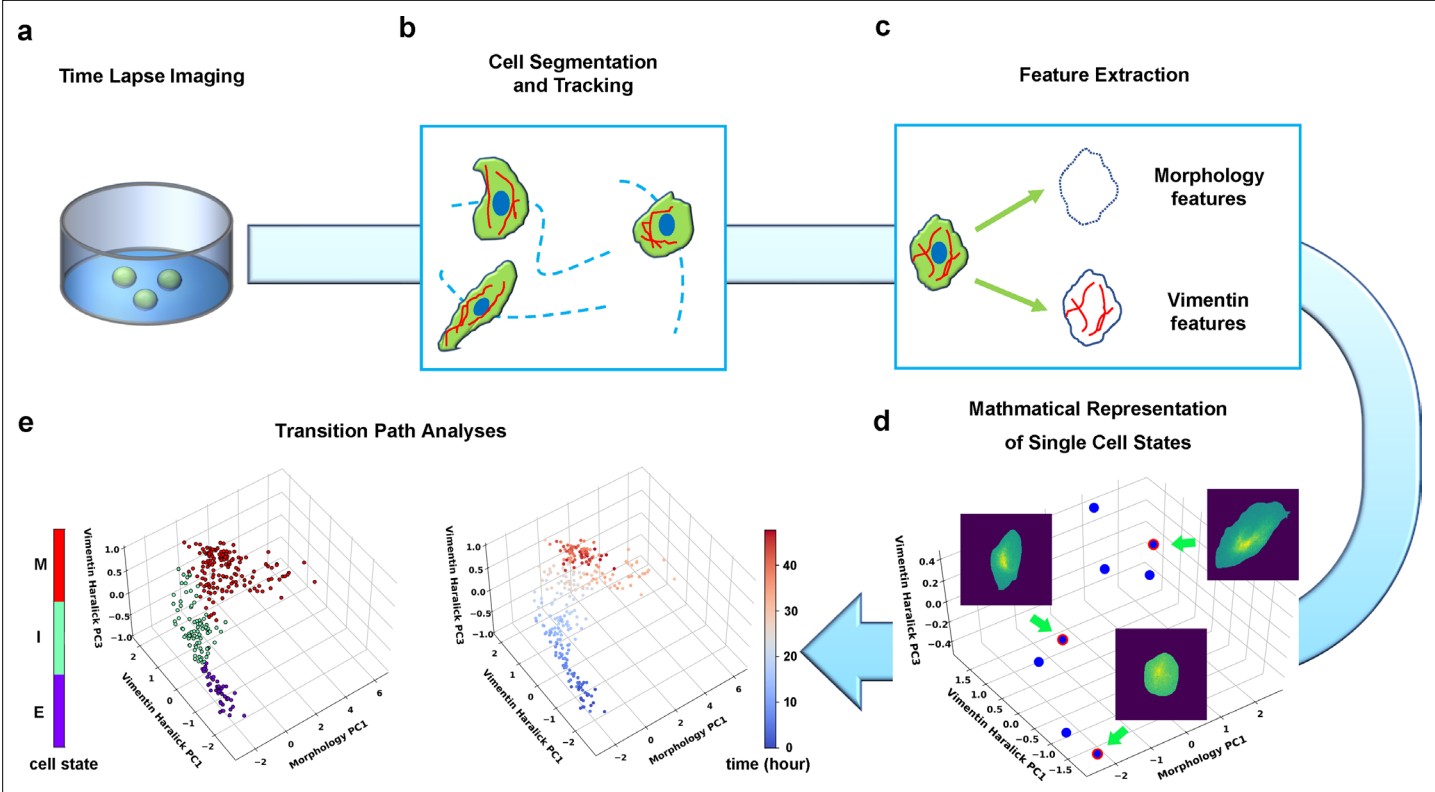

**Figure 2.** Summary of pipeline for recording and analyzing single-cell trajectories in composite multi-dimensional cell feature space. (**a**) Time-lapse imaging of A549/Vim-RFP cells treated with TGF-β. (**b**) Deep-learning aided single-cell segmentation and tracking on the acquired time-lapse images. (**c**) Extraction of morphology and vimentin features of single cells. (**d**) Representation of single cell states in a multidimensional morphology/texture feature space. (**e**) Transition path analyses over recorded trajectories. Right: A representative single cell trajectory of EMT in the feature space. Color represents time (unit: hour). Left: the same trajectory colored by the regions in the feature space (E, I, and M, for epithelial, intermediate, and mesenchymal regions, respectively) each data point resides. Reduced units are used in this and all other figures.

(*Gordonov et al., 2016*; *Chang and Marshall, 2019*; *Kimmel et al., 2018*). Noticing that cells have phenotype-specific morphological features that can be monitored even with transmission light microscopes, recently we developed a framework that defines cell states in a combined morphology and texture feature space (*Wang et al., 2020*). The framework allows one to trace individual cell trajectories during a CPT process through live-cell imaging. We applied the framework to study the TGF-β induced EMT with the A549/Vim-RFP cells (*Figure 2*; *Wang et al., 2020*). Vimentin is a type of intermediate filament protein commonly used as a mesenchymal marker, and changes its expression and spatial distribution during EMT (*Zhang et al., 2014*). Furthermore, during EMT cells undergo dramatic change of cell morphology accompanying gene expression change (*Figure 1a*; *Zhang et al., 2014*; *Zhang et al., 2019*). Accordingly, we used a combination of vimentin texture and cell shape features to specify a cell state.

We first performed time-lapse imaging of the EMT process of A549/Vim-RFP induced with 4 ng/ml TGF-β (*Figure 2a*, Materials and methods *Wagner and Klein, 2020*; *Weinreb et al., 2020*), then performed single-cell segmentation and tracking on the acquired images (*Figure 2b*, Materials and methods *Huang et al., 2005*). We quantified the images with an active shape model (*Cootes et al., 1995*) and performed principal component analysis (PCA) to form a set of orthonormal basis vectors of collective variables, which include cell body shape of 296 degrees of freedom (DoF) quantified by an active shape model (*Cootes et al., 1995*), and texture features of cellular vimentin distribution quantified by 13 Haralick features (*Haralick, 1979*; *Figure 2c*, Materials and methods *Huang et al., 2005*). Then the state of a cell at a given instant is represented as a point in the 309-dimensional composite morphology/texture feature space (*Figure 2d*), and the temporal evolution of the state forms a continuous trajectory in the space subject to further theoretical analyses (*Figure 2e*).

## Transition path analyses identify an ensemble of reactive single-cell trajectories

Before TGF-β treatment, a population of cells assume a localized stationary distribution in this 309-dimensional composite space, and most cells are epithelial (*Figure 3a*, blue). TGF-β treatment destabilizes such distribution, and the cells relax into a new stationary distribution dominated by mesenchymal cells (*Figure 3a* red). We recorded 204 continuous trajectories in the state space. A representative trajectory shown in *Figure 2e* (right) (also in *Video 1*) reveals how a cell transits step-by-step from an epithelial cell with convex polygon shapes and a localized intracellular vimentin distribution, to the mesenchymal phenotype with elongated spear shapes and a dispersive vimentin distribution.

Next, we applied rate theory analyses on the recorded single-cell trajectories. Rate theory studies how a system escapes from a metastable state, or relaxation from one stationary distribution to a new one (*Hänggi et al., 1990*). Specifically, transition path theory (TPT) is a modern development in the field (*Weinan et al., 2005*; *Rohrdanz et al., 2013*; *Bolhuis et al., 2002*; *Weinan and Vanden-Eijnden, 2010*). Within the TPT framework, one divides the state space into regions containing the initial (*A*) and final (*B*) attractors, and an intermediate (*I*) region. A reactive trajectory is one that originates from region *A*, and enters region *I* then *B* before re-entering region *A* (*Figure 3b*). The reactive trajectories form an ensemble of transition paths that connect regions *A* and *B*.

Accordingly, we divided the four-dimensional principal component subspace into epithelial (*E*), intermediate (*I*), and mesenchymal (*M*) regions (*Figure 2e* left) (*Wang et al., 2020*). With this division of space, we identified a subgroup of 135 recorded single-cell trajectories that form an ensemble of reactive trajectories that connect *E* and *M* by day 2. It should be noted that the practical definition of reactive trajectories here differs from the theoretical definition in classical TPT. In classical TPT, one runs a trajectory of infinite length that travel back and forth between the initial and final regions numerous times, and a reactive trajectory refers to a segment that starts from the initial region and ends in the final region.

## TGF-β induced EMT in A549/Vim-RFP cells proceeds through parallel transition paths

We first examined whether EMT proceeds through one or multiple types of paths by analyzing the ensemble of reactive trajectories using self-organizing map (SOM) (Materials and methods *Rand et al., 2021*). SOM is an unsupervised artificial neural network that utilizes a neighborhood function to represent the topology structure of input data (*Kohonen, 1982*). The algorithm clusters the recorded cell states into 144 discrete states, and represents the EMT process as a Markovian transition process among these states (*Figure 3—figure supplement 1*). Shortest path analysis using the Dijkstra algorithm (*Dijkstra, 1959*) over the transition matrix reveals two groups of paths: vimentin Haralick PC1 varies first, and concerted variation of morphology PC1 and vimentin Haralick PC1, with finite probabilities of transition between the two groups (*Figure 3c*). This result is consistent with our previous trajectory clustering analysis using dynamics time warping (DTW) distance between reactive trajectories (*Wang et al., 2020*). To further support this conclusion, we examined the density of reactive trajectories $\rho_R$ in the plane of morphology PC1 and vimentin Haralick PC1 (Materials and methods *Mojtahedi et al., 2016*). The contour map of $\rho_R$ shows two peaks corresponding to the two groups of shorted paths in the directed network (*Figure 3d*). The peak where vimentin Haralick PC1 varies first is higher than the peak of concerted variation, indicating more reactive trajectories along this path. The distinct features of the two types of trajectories are apparent from the two representative single-cell trajectories shown in *Figure 3e*.

## A revised string method is developed to reconstruct reaction coordinate from single-cell trajectories

Due to the continuous and stochastic nature of the system dynamics there are infinite number of reactive trajectories, which mainly concentrate within a tube in the state space that connects regions *A* and *B* (*Figure 3b*; *Weinan and Vanden-Eijnden, 2010*). The center of the tube has been used to define a reaction coordinate (RC), a concept central to rate theories (*Figure 4a*; *Vanden-Eijnden and Venturoli, 2009*). A RC is a one-dimensional geometric parameter (denoted as *s* in the subsequent discussions) that describes the progression along a continuous reaction path defined in the state

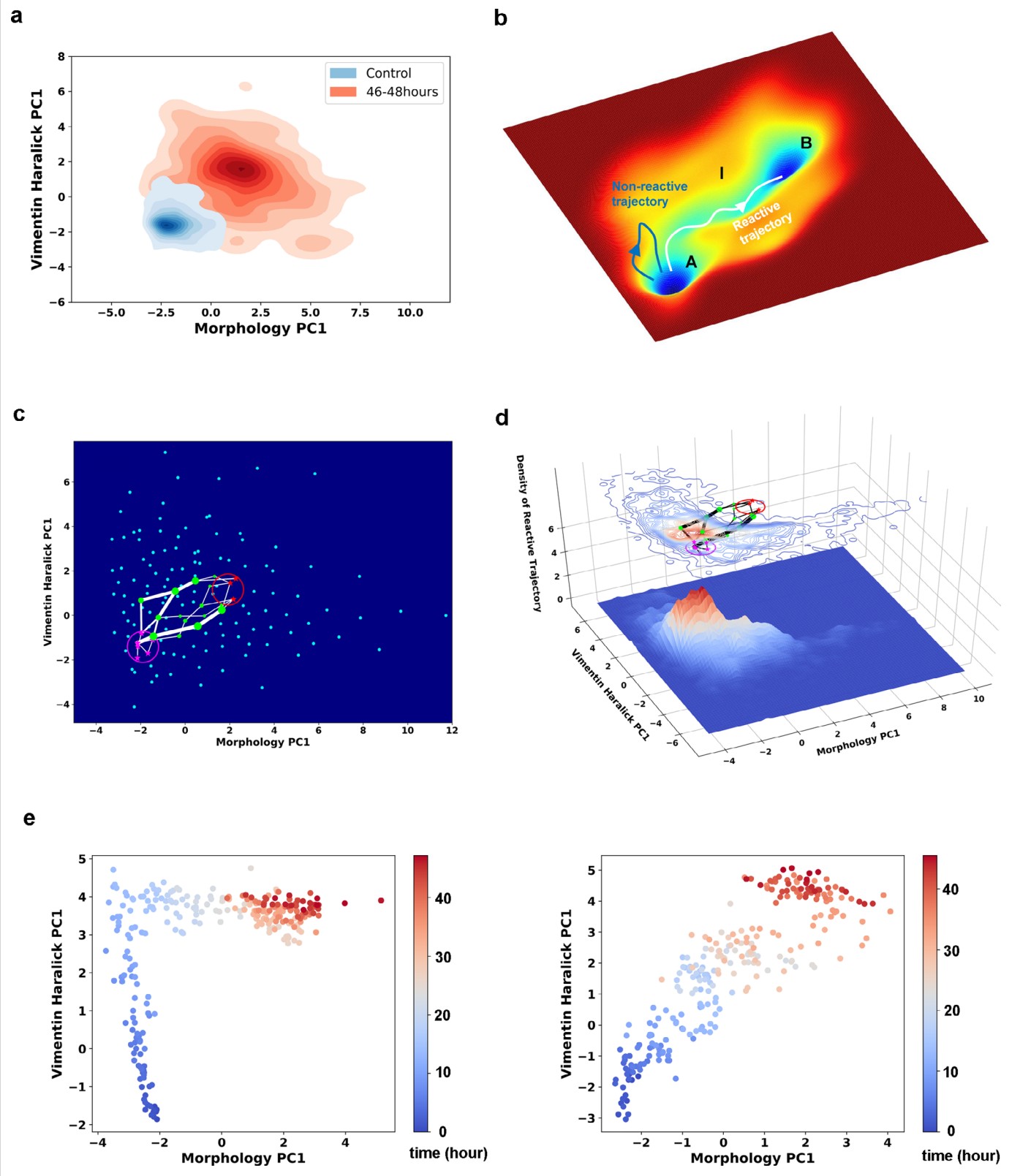

**Figure 3.** Single cell trajectories of EMT form two distinct groups. (**a**) Kernel density estimation of cells in control condition (Blue) and cells after being treated with 4 ng/ml TGF-β for 46–48 hours (Red). (**b**) Schematic example reactive and nonreactive trajectories in a potential system. Also shown is a valley connecting regions A and B that most reactive trajectories fall in and form a reaction tube. (**c**) Discretization of the whole single cell data set with self-organizing map into 12 × 12 discrete states (clusters) in the state space. Directed network generated base on the self-organizing map and

*Figure 3 continued*

the transitions between states. The distance between two states is defined as the negative logarithm of transition probability. White lines are shortest between epithelial states (purple stars) and mesenchymal states (red stars). Green dots are states that the shorted paths pass by. The size of a dot stands for the frequency of this dot passed by shortest paths. The width of a white line represents the frequency that these shortest paths pass by. (**d**) Contour map (top, superimposed with the shortest paths in panel **c**) and 3D surface-plot (bottom) of density of reactive trajectories in the plane of morphology PC1 and vimentin Haralick PC1. (**e**) Representative trajectories from the two groups. Left: Vimentin varies first. Right: Concerted variation.

The online version of this article includes the following figure supplement(s) for figure 3:

**Figure supplement 1.** Space approximation of the whole single cell data set with self-organizing map into 12 × 12 discrete states (clusters).

space (*Rohrdanz et al., 2013*). A good choice of RC can provide mechanistic insight on how the transition process proceeds.

Therefore, we set to reconstruct the RC for each path with the recorded reactive trajectories using a revised finite temperature string method, which was originally designed for RC reconstruction for theoretical and computational modeling systems (*Vanden-Eijnden and Venturoli, 2009*; *Allen et al., 2005*; *Dickson et al., 2009*). The method first approximates the RC by a set of discrete image points that are uniformly distributed along the arc length of the RC, so the multi-dimensional state space can be divided by a 1-D array of Voronoi polyhedra containing individual images, and the data points can be assigned to individual Voronoi cells (*Figure 4a*). Starting with a trial RC to define the initial division of the Voronoi cells, one optimizes the trial RC iteratively by minimizing the distance dispersion between the string point and sample points within each Voronoi cell (*Figure 4—figure supplement 1*). Since here we have an ensemble of continuous trajectories, we modified the iteration procedure slightly. Specifically, we minimized both the distance between the ensemble of measured reactive trajectories and the image point within each individual Voronoi cell, as well as the overall distance between each individual trajectory and the trial RC (Materials and methods *Chen et al., 2012*).

Therefore, we grouped the reactive trajectories based on the DTW distance, then identified the RCs for each group separately following the modified string method. The iteration procedure gives the RC of each path ($s_1$ and $s_2$) that characterizes common features of the reactive trajectories for the TGF-β induced EMT in A549/Vim-RFP cells (*Figure 4b*). The recorded trajectories fluctuate around the RCs and form reaction tubes as expected from the TPT theory (*Figure 4c*). Along each RC, the cell shape changes dominantly through elongation and growth (*Figure 4d & e*, left), and most of the 13 vimentin Haralick features increase or decrease monotonically and continuously over time (*Figure 4d & e*, right) (Materials and Methods *Tian et al., 2013*). The two RCs first diverge from the *E* region to follow two distinct paths, then converge within the *M* region. In one group (*Figure 4d*), most of the Haralick feature changes take place before major morphology change. For the group with concerted dynamics (*Figure 4e*), both cell shape and Haralick features vary gradually along the RC.

## Reconstructed reaction coordinate and quasi-potentials reveal EMT as relaxation along continuum manifolds

Considering a cell as a dynamical system, with the single-cell trajectories one may formulate an inverse problem to reconstruct the underlying dynamical equations governing the EMT transition. Let's take a minimal ansatz that assumes the dynamics of the collective variables (**x**) can be described by a set of Langevin equations in the morphology/texture feature space, $d\mathbf{x}/dt = \mathbf{F}(\mathbf{x}) + \eta(\mathbf{x}, t)$, where **F**(**x**) is a governing vector field, and $\eta$ are white Gaussian noises with zero mean. Then in principle one can reconstruct **F**(**x**) from the single-cell trajectory data, $\mathrm{F}(x) = \langle \mathrm{d}x/\mathrm{d}t \rangle$, averaged over the neighborhood of each **x**. Notice for the ensemble average one needs to use all the reactive and nonreactive trajectories.

For mathematical simplicity and better numerical convergence, here we restricted to reconstructing the dynamics along the RC (*Figure 5a*).

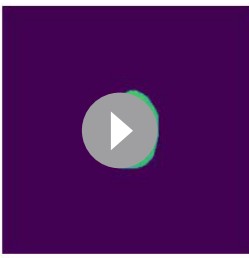

**Video 1.** Recorded live cell trajectory in Figure 2e. Each frame is a segmented cell mask cropped from the original vimentin fluorescence image.
https://elifesciences.org/articles/74866/figures#video1

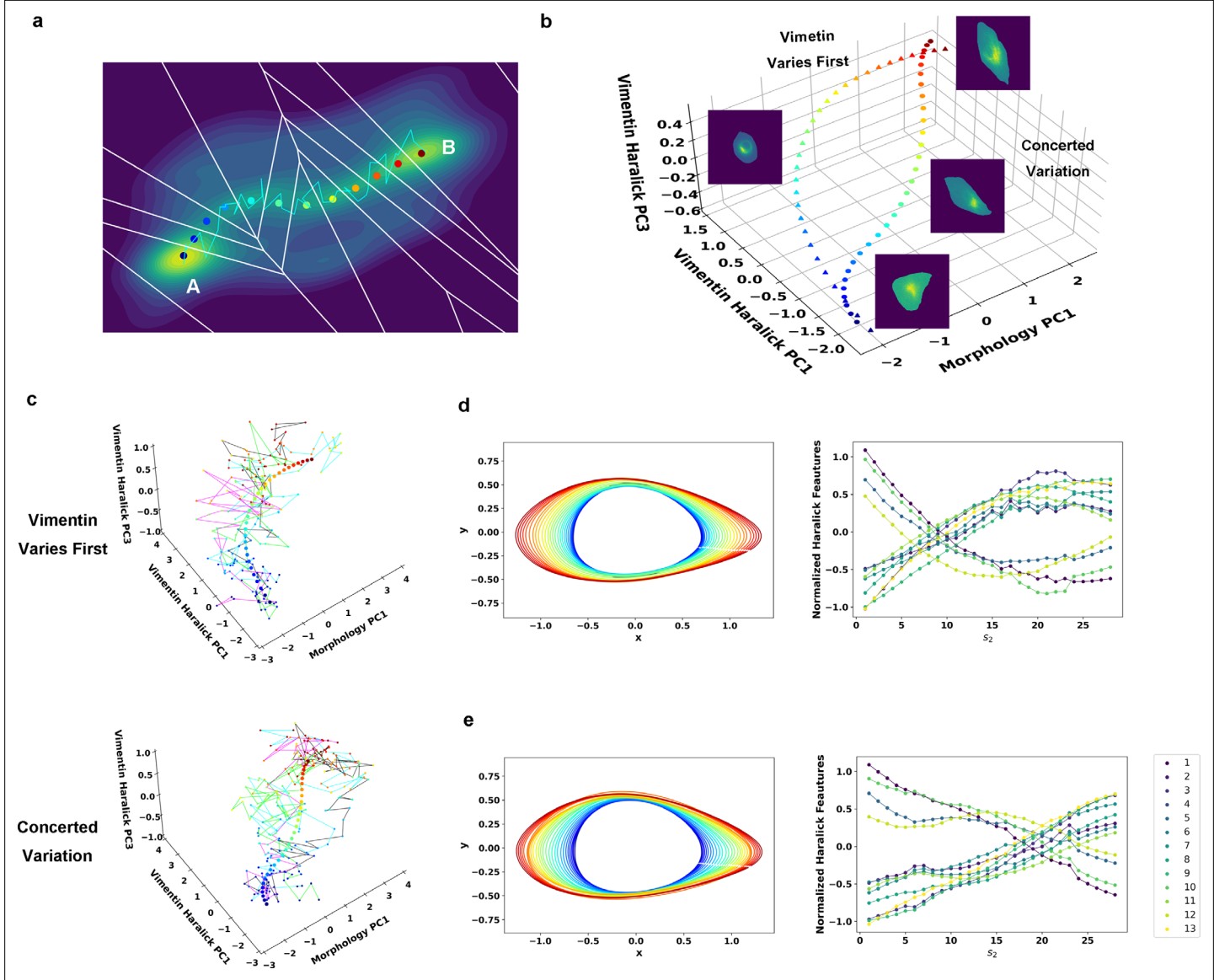

**Figure 4.** Reaction coordinate reconstruction of two parallel paths from reactive A549/Vim-RFP EMT trajectory ensemble with a modified string method. (**a**) Discrete representation of a 1-D reaction coordinate (i.e. image points, represented as colored dots) on the filled contour map with corresponding Voronoi cells. The cyan line is a reactive trajectory that starts from A and ends in B. (**b**) Reconstructed RCs from reactive single cell EMT trajectories using a revised finite temperature string method. Single cell images represent typical cell states in their locations (Epithelial state, mesenchymal state, in mid-transition with vimentin varies first and in mid-transition with concerted variation) (**c**) Separate representation of the RCs overlaid with representative reactive single-cell trajectories. Top: vimentin varies first; Bottom: concerted variation. For visual clarity, the trajectories were plotted with data points separated by 50 min. (**d**) Cell shape (left), Haralick feature (right) along RC $s_1$. The colors of the cell shapes in b/c left represent progression of EMT (starts from blue and ends in red). Haralick feature 1: Angular Second Moment; 2: Contrast; 3: Correlation; 4: Sum of Squares: Variance; 5: Inverse Difference Moment; 6: Sum Average; 7: Sum Variance; 8: Sum Entropy; 9: Entropy; 10: Difference Variance; 11: Difference Entropy; 12: Information Measure of Correlation 1; 13: Information Measure of Correlation 2. (**e**) Similar to panel e but along RC $s_2$.

The online version of this article includes the following figure supplement(s) for figure 4:

**Figure supplement 1.** Iterative procedure of the finite temperature string method.

The ansatz becomes a 1-D convection-diffusion process, $ds/dt = -d\phi/ds + \eta$. Notice that for a 1-D system even without detailed balance one can define a quasi-scalar potential $\phi$ (**Xing, 2010**; **Xing and Kim, 2011**; **Risken, 1996**). This quasi-potential corresponds to a potential of mean force in the case of a conserved system. To confirm that the ansatz of using a memory-less 1-D Langevin equation is a good approximation for the EMT dynamics along a RC, we performed the Chapman-Kolmogorov

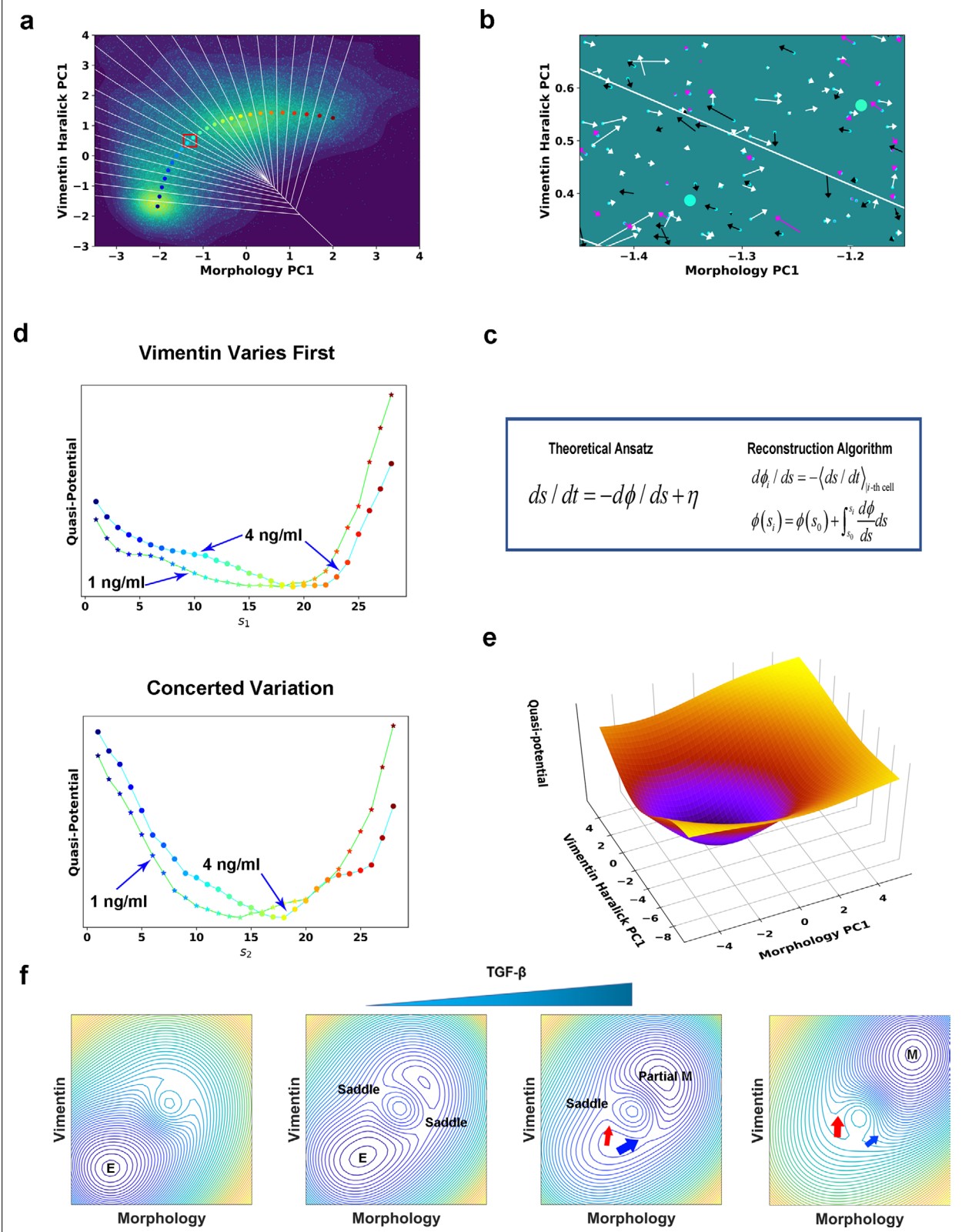

**Figure 5.** Quantification of dynamics along the two RCs suggests a mechanism of forming two EMT paths through sequential saddle-node collisions. (**a**) Reconstructed RC $s_1$ on density plot of single cell trajectories data. Cyan dots are single cell data points. (**b**) Enlarged view of the red box region in (**a**). The arrow associated with each data point (cyan dot) represents the value of ds/dt (white: > 0, magenta: = 0, black: < 0). (**c**) Theoretical framework of dynamics reconstruction along the RC. (**d**) Comparison of reconstructed quasi-potentials along the RC with 1 ng/ml (star) and 4 ng/ml (circle) TGF-β

*Figure 5 continued*

treatment. Top: Reconstructed quasi-potentials along RC $s_1$. Bottom: Reconstructed quasi-potentials along RC $s_2$. (e) Quasi-potential of the control cells based on kernel density estimation. (f) A metaphorical potential system to illustrate a plausible mechanism for generating the two paths through sequential collision between a stable attractor and two saddle points when the concentration of TGF-β increases. The width of the arrows represents the probabilities that single-cell trajectories follow corresponding paths. E: epithelial attractor; Partial M: partial EMT attractor; M: mesenchymal attractor.

The online version of this article includes the following figure supplement(s) for figure 5:

**Figure supplement 1.** Chapman-Kolmogorow test on the RC $s_1$ (**a**) and $s_2$ (**b**).

**Figure supplement 2.** Reconstructed $d\Phi/ds$ along the RC in the case of 4 ng/ml TGF-β treatment.

**Figure supplement 3.** Comparison between the predicted steady distribution of RC values from reconstructed potential and experiment data.

**Figure supplement 4.** Embedding of dynamics time warping (DTW) distance matrix of reactive trajectories under 1 ng/ml (**a**) and 4 ng/ml (**b**) TGF-β treatment with t-SNE.

**Figure supplement 5.** Additional analyses of trajectories with 1 ng/ml TGF-β treatment.

**Figure supplement 6.** Reconstructed potentials of different numbers of points in the reaction coordinates Comparison of reconstructed quasi-potentials along the RC with 1 ng/ml (star, lime colored line) and 4 ng/ml (circle, cyan colored line) TGF-β treatment.

**Figure supplement 7.** Pipeline of singe-cell trajectory analyses.

Test (CK-test). The test shows that the one-step transition matrix can indeed predict the dynamics on longer time scales, and thus the Markovian assumption is a good zero[th]-order approximation of the EMT dynamics (*Figure 5—figure supplement 1*, Materials and Methods *Yang et al., 2020*).

To reconstruct the dynamical equations, we used the measured trajectories and instant velocities (along the RC direction) as input. An enlarged view in *Figure 5b* shows the instant velocities ($ds/dt$) of various trajectories segments within Voronoi cells. Numerically, we related the potential gradient $d\phi/ds$ with the mean velocity within the $i_{th}$ Voronoi cell through averaging over all recorded reactive and nonreactive trajectory segments that locate within the Voronoi cell at any time $t$ (*Figure 5a & c*, Materials and methods *McFaline-Figueroa et al., 2019*; *Nieto et al., 2016*). On the obtained curve of $d\phi/ds$ vs. $s$ (*Figure 5—figure supplement 2*), the zeroes correspond to stationary points of the potential. We then reconstructed the quasi-potential through integrating over $s$, $\phi(s_i) = \phi(s_0) + \int_{s_0}^{s_i} (d\phi/ds)ds$ (*Figure 5d*). We also obtained the quasi-potential of the untreated cells (*Figure 5e*) from the steady state distribution of untreated cells along the RC, $\phi_0 \propto -log p_{ss}$.

The reconstructed RCs and quasi-potentials provide mechanistic insight on the transition. Before TGF-β induction, cells reside on the untreated cell potential centered with a potential well corresponding to the epithelial attractor (*Figure 5e*). After induction, the system relaxes following the new potential into a new well corresponding to the mesenchymal attractor. Notably, in the new potential the original epithelial attractor disappears, reflecting that the epithelial phenotype is destabilized under the applied 4 ng/ml TGF-β concentration.

With the reconstructed quasi-potentials and variance of $d\phi/ds$ obtained from experiment, we solved the Fokker-Planck (FP) equation corresponding to the Langevin equation to predict the steady distribution vs. RC (Materials and methods *Wang et al., 2020*). While the dynamical equations were reconstructed from trajectories during the transition process and with a minimal Langevin equation ansatz, the predicted stationary distribution have average values and standard deviations, (17.2, 4.6) and (14.7, 4.4) for the two paths, respectively, close to those calculated from the distributions sampled from cells after 39 h TGF-β treatment, (16.0, 5.0) and (16.2, 4.0), respectively (*Figure 5—figure supplement 3a and b*).

## Lowering TGF-β concentration leads to relaxation to a new partial EMT attractor through two parallel paths

Notice that quasi-potential with 4 ng/ml TGF-β for the Vimentin varies first path shows a plateau around $s_1 = 8$ (*Figure 5d*). We hypothesized that this plateau is a remnant of the original epithelial attractor, and if so we expect to observe a flatter plateau or even a metastable attractor with a lower TGF-β concentration. Therefore, we treated A549/Vim-RFP cells with 1 ng/ml TGF-β and recorded 135 3-day long single-cell trajectories. Among the recorded trajectories 65 indeed reached the *M* region. The reactive trajectories also cluster into two groups (*Figure 5—figure supplement 4*). For the purpose of comparison, we projected the trajectories onto the RCs obtained from the 4 ng/ml TGF-β and reconstructed the quasi-potentials in the case of 1 ng/ml of TGF-β. Indeed, the quasi-potential for

the path with vimentin varying first has a plateau flatter than that of the 4 ng/ml TGF-β-treated cells. The quasi-potential of another path, however, does not show such flattening.

In the EMT field, it is under debate whether EMT has a discrete or continuum spectrum of intermediate states (*Yang et al., 2020*). Single-cell transcriptomic studies reveal intermediate states (*Cook and Vanderhyden, 2020*). If EMT proceeds through a set of discrete and definite states, we expect to observe attractors, i.e., potential wells, corresponding to these states. In addition, lowering the TGF-β concentration would lead to an increased barrier between neighboring wells and cells trapped at some intermediate state for a long time. While all the quasi-potentials at the two TGF-β concentrations reveal destabilization of the epithelial attractor, they do not have multiple attractors anticipated for discrete EMT states (*Figure 5e $d\phi/ds$* in *Figure 5—figure supplement 5a, b*). Compared to those at 4 ng/ml TGF-β, both quasi-potentials at 1 ng/ml TGF-β have the new attractor closer to the epithelial state. Examination of individual trajectories also reveals that cells with 1 ng/ml TGF-β treatment leave the *E* region and mostly reside around a new attractor in the *I* region (*Figure 5—figure supplement 5c, d*). Some trajectories fluctuate within the attractor and only transiently reach the *M* region. That is, cells reach a new stable phenotype whose degree of mesenchymal features depends on the TGF-β concentration, which is consistent with previous studies on MCF10A cells treated with different concentrations of TGF-β (*Zhang et al., 2014*). Fluctuations of single-cell trajectories limit the state resolution in the cell feature space and prevent us from telling whether the two paths reach one common intermediate state or two distinct ones. To examine how the number of Voronoi cells affects the quasi-potential properties, we varied the number of discretization points along the RCs and did the same analysis to reconstruct the quasi-potential (*Figure 5—figure supplement 6a and b*). The reconstructed potentials have similar shapes, although show different smoothness.

## Discussion

Cell type regulation is an important topic in mathematical and systems biology, and several theoretical and computational studies on modeling CPT systems have been performed in the context of rate theories (*Aurell and Sneppen, 2002*; *Brackston et al., 2018*; *Morelli et al., 2008*; *Wang et al., 2014*; *Ge et al., 2015*; *Wang et al., 2011*; *Tse et al., 2018*). Recent advances in single-cell techniques have catalyzed quantitative measurements on the dynamics of CPTs, and have opened new questions on how to quantify and analyze single-cell data in the framework of dynamical systems theory and how it compares with theoretical results. In our previous studies (*Wang et al., 2020*), we developed an integrated live cell imaging and image analysis framework for quantifying and representing live cell imaging data. In this work, we further showed that the mathematical representation allows one to analyze the single cell trajectories and study CPT dynamics using rate theories for quantitative mechanistic insights (*Figure 5—figure supplement 7*).

A long-held concept in cell and developmental biology is that cells exist as discrete and distinct phenotypes. Single cell genomics measurements have challenged such views and have revealed that cells move along continuum manifolds. The live cell imaging studies presented here support such a picture by demonstrating EMT as a relaxation along a continuum manifold, which is also supported by single cell RNA-seq studies (*McFaline-Figueroa et al., 2019*).The high-throughput single-cell data provides high dimensional information, but identifying the intermediate states in the snap-shot data requires the development of new dimension reduction and clustering methods (*MacLean et al., 2018*). Compared to the snapshot studies, live cell trajectories reveal that the EMT trajectories can be clustered as fluctuating around two distinct paths that connect the epithelial and mesenchymal phenotypes. The existence of two types of transition paths suggest that the previously proposed conceptual 1D EMT axis (*Nieto et al., 2016*) is insufficient for understanding how EMT proceeds, and puts dynamics inferred indirectly from snapshot data questionable (*McFaline-Figueroa et al., 2019*). Extracting multi-dimensional features from live cell imaging data provides a complementary strategy, and further development along this direction will benefit from advances of imaging technology and algorithm (*Christiansen et al., 2018*; *Ounkomol et al., 2018*).

In rate theories, identifying an appropriate RC provides mechanistic insights of a transition process. For example, recognizing collective solvent reorganization as part of the reaction coordinate is a key part of Marcus's electron transfer theory (*Marcus, 1993*). Adopting the fraction of native contacts as a RC also historically plays a key role in the development of protein folding theories. Therefore, continuous efforts have been made on defining an optimal RC for a complex dynamical system (*Li*

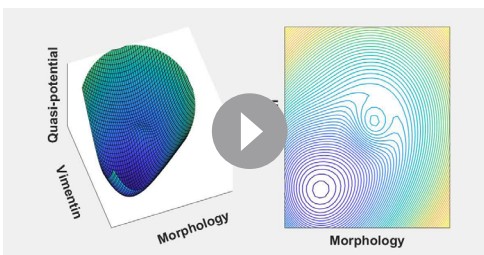

**Video 2.** A metaphorical potential system illustrating how TGF-β treatment modifies the cell dynamics. https://elifesciences.org/articles/74866/figures#video2

and Ma, 2014; Best and Hummer, 2005; Fukui, 2002), and it is even questionable to assume that the system dynamics can be well presented by a 1-D RC, as suggested by theoretical analysis on protein folding (Udgaonkar, 2008). Indeed, the analyses presented in this study shows such a breakdown of this basic assumption.

In the introduction, we discussed two basic mechanisms of critical state transitions. For TGF-β induced EMT in A549 cells, analysis of the trajectories reveals that mathematically multiple paths may originate from destabilization of a multi-dimensional epithelial attractor through colliding with multiple saddle points sequentially.

We illustrated such a mechanism in *Figure 5f* and in *Video 2* schematically with a metaphorical potential system. With no or low TGF-β, the system resides in the epithelial attractor (*Figure 5f*, first). Adding TGF-β leads to the appearance of a new (mesenchymal) attractor, and elevation of the epithelial attractor (*Figure 5f*, second). The epithelial attractor approaches and collides first with a saddle point to form a barrier-less (concerted-variation) path to the new attractor (*Figure 5f*, third). At this TGF-β concentration (e.g. 1 ng/ml), some barrier still exists along an alternative (vimentin-first) path, which then disappears with the increase of TGF-β concentration again (as revealed in *Figure 5d*) through saddle node collision (*Figure 5f*, fourth). Indeed, we observed the reactive trajectories predominantly assume the concerted-variation path under 1 ng/ml TGF-β treatment (*Figure 5—figure supplement 4a*), while the vimentin-first path becomes dominant under 4 ng/ml TGF-β treatment (*Figure 5—figure supplement 4b*). It should be pointed out that this sequential saddle-node bifurcation mechanism is only a plausible one that can explain the experimental data. Multidimensional systems can show complex bifurcation patterns (*Rand et al., 2021*; *Kheir Gouda et al., 2019*). With sufficient single-cell trajectories, one can extend the procedure described in this work and in *Qiu et al., 2022* to reconstruct the multi-dimensional vector field and the full governing Fokker-Planck equations directly. The full model will help address questions on the role of noise in CPTs (*Balázsi et al., 2011*).

One major limitation of the minimal Langevin equation ansatz used in this work is that the dynamics is assumed to be Markovian, that is, the temporal evolution of the system depends only on the current but not previous cell state. A cell has a much larger number of degrees of freedom than what can be measured explicitly through live-cell imaging, and these implicit degrees of freedom result in possible non-Markovian dynamics. Future studies may use the generalized Langevin equation formalism that takes into account the non-Markovian effects. In addition, we used the added exogenous TGF-β concentration as a control parameter, as in previous studies (*Tian et al., 2013*; *Zhang et al., 2014*). Cells also secrete endogenous TGF-β, and interact with other cells, so the extracellular microenvironment for individual cells is in general spatially and temporally changing. In future studies, one can relax the mean-field approximation assumed in this work. Experimentally, one can use microfluidic chambers for more precise control of extracellular environments. Furthermore, one needs to identify the expression profiles of cell states along the two paths for deeper mechanistic understanding how the two paths emerge and how one can modulate the EMT dynamics, possibly through combining live cell imaging and fixed-cell measurements.

In summary, through an integrated framework of live cell imaging and single-cell trajectories with dynamical systems theories we obtained quantitative mechanistic insights of TGF-β induced EMT as a prototype for CPTs in general. Since a cell is a complex dynamical system with many strongly coupled degrees of freedom and a broad range of relevant time scales, further development will benefit from a finer resolution of cell state through including additional measurable cell features.

# Materials and methods

**Key resources table**

| Reagent type (species) or resource | Designation | Source or reference | Identifiers | Additional information |
|---|---|---|---|---|
| Cell line (*Homo sapiens*, human) | A-549 VIM RFP | American Type Culture Collection(ATCC) | Cat# CCL-185EMT RRID:SCR_007358 | |
| Peptide, recombinant protein | TGF-β (Recombinant Human TGF-beta 1 Protein) | R&D Systems | Cat#240-B | |
| Software, algorithm | CellProfiler | Broad Institute | RRID:SCR_007358 | |
| Software, algorithm | Scipy | PMID:32015543 | RRID:SCR_008058 | https://scipy.org/ |

## Cell line

A549/VIM-RFP cells were obtained from (ATCC CCL-185EMT, RRID:CVCL_LI35), and cells within 5–15 generations were used in the studies. Cells were cultured in F-12K medium (Corning) with 10% fetal bovine serum (FBS) in MatTek glass bottom culture dishes (P35G-0–10 C) in a humidified atmosphere at 37 °C and 5% $CO_2$, as detailed in *Wang et al., 2020*. The cell line was authenticated as A549 cells with short tandem repeat (STR) analysis by University of Arizona Genetics Core (UAGC). The result of mycoplasma contamination test (MycoFluor Mycoplasma Detection Kit Molecular Probes, M-7006) was negative.

## Time-lapse imaging

Time-lapse images were taken with a Nikon Ti2-E microscope with differential interference contrast (DIC) and TRITC channels (Excitation wavelength is 555 nm and Emission wavelength is 587) (20× objective, N.A. = 0.75). The cell culture condition was maintained with Tokai Hit Microscope Stage Top Incubator. For the 4 ng/ml TGF-β (R&D Systems 240-B) experiment, cells were imaged every 5 min with the DIC channel and every 10 min with the TRITC channel for 2 days. The exposure time for DIC was 100 ms and the exposure time for the TRITC channel was 30 ms. For the 1 ng/ml TGF-β experiment, cells were imaged every 5 min with the DIC channel and every 15 min with the TRITC channel for 3 days. The exposure time for DIC was 100ms and the exposure time for the TRITC channel was 30ms. While taking the images, all the imaging fields were chosen randomly.

## Trajectory analyses

After single-cell segmentation, we performed single cell tracking with CellProfiler (RRID:SCR_007358) using a linear assignment problem (LAP) method (*Jaqaman et al., 2008*; *Carpenter et al., 2006*). To reduce the influence of over- and under-segmentation on the trajectory analysis, we utilized the cell tracking to detect these false segmentations through calculating the overlap relationship between single cells in the consecutive frames. If a cell in one frame and its maximum overlap cell in the next frame do not belong to the same trajectory, cells that overlap with the two cells were manually checked. Specifically, we trained an CNN (ResNet) to identify the over- and under-segmented cells (*He et al., 2016*). To increase accuracy, we used both the prediction results from this CNN, and other criteria. For instance, falsely segmented cells usually do not exist for a long duration. We used the lasting duration (30 minutes) to identify possible over- and under-segmentation. The identified falsely segmented cells were removed from the trajectories, and the broken trajectories were relinked using the LAP method.

We represented the state of a cell by its morphology and Vimentin Haralick features (*Wang et al., 2020*). The morphology was described with the active shape model (*Pincus and Theriot, 2007*). Calculation of Haralick features was based on the gray-level co-occurrence matrix G. $G_{ij}$ is the frequency of observing the gray-levels values of two neighboring pixels i and j, respectively. It was normalized by the total counts. In 2D images, there are four directions of such neighbors (0°, 45°, 90°, and 135°). To be rotation invariant, each feature was calculated on all the four directions of G matrix and averaged over them. Haralick features include the following features: 1 Angular Second Moment; 2: Contrast; 3: Correlation; 4: Sum of Squares: Variance; 5: Inverse Difference Moment; 6: Sum Average; 7: Sum Variance; 8: Sum Entropy; 9: Entropy; 10: Difference Variance; 11: Difference Entropy; 12: Information Measure of Correlation 1; 13: Information Measure of Correlation 2.

## Self-organizing map and shortest transition paths in the directed network

The self-organizing map is an unsupervised machine learning method to represent the topology structure of date sets. We used a 12 × 12 grid (neurons) to perform space approximation of all reactive trajectories. The SOM was trained for 50 epochs on the data with Neupy (http://neupy.com/pages/home.html). We set the learning radius as one and standard deviation 1. These neurons divide the data into 144 micro-clusters ($\{\psi\}$). With the single-cell trajectory data, we counted the transition probabilities from cluster $i$ to cluster $j$ (including self), with $\sum_j p_{ij} = 1$. If the transition probability is smaller than 0.01, the value is then reset as 0. With the transition probability matrix, we built a directed network of these 144 neurons (*Figure 3—figure supplement 1*). The distance (weight) of the edge between neuron $\psi_i$ and neuron $\psi_j$ is defined as the negative logarithm of transition probability ($-\log p_{ij}$). We set the neurons that close to the center of epithelial and mesenchymal state (sphere with radius = 0.7) as epithelial community and mesenchymal community, respectively, and used Dijkstra algorithm to find the shortest path between each pair of epithelial and mesenchymal neurons (*Dijkstra, 1959*) with NetworkX (*Hagberg et al., 2008*). We recorded the frequency of neurons and edge between these neurons that were past by these shortest paths.

## Calculation of density of reactive trajectories

The density of reactive trajectory on the plane of morphology PC1 and vimentin Haralick PC1 was calculated with the following procedure:

a. Divide the whole plane into 200 × 200 grids.
b. In each grid, count the number of reactive trajectory (only the parts of each reactive trajectory that were in the intermediate region were taken into consideration) that enters and leaves it. If a reactive trajectory passes certain grid multiple times, only one was added in this grid's density. Thus, the density matrix was obtained.
c. Use Gaussian filter to smooth the obtained density matrix. The standard deviation was set to be two and the truncation was two (i.e., truncate at twice of the standard deviation).

## Procedure for determining a reaction coordinate

We followed a procedure adapted from what used in the finite temperature string method for numerical searching of reaction coordinate and non-equilibrium umbrella sampling (*Vanden-Eijnden and Venturoli, 2009*; *Dickson et al., 2009*), with a major difference that we used experimentally measured single-cell trajectories (*Figure 4—figure supplement 1*).

a. Identify the starting and ending points of the reaction path as the means of data points in the epithelial and mesenchymal regions, respectively. The two points are fixed in the remaining iterations.
b. Construct an initial guess of the reaction path that connects the two ending points in the feature space through linear interpolation. Discretize the path with $N$ ( = 30) points (called images, and the $k_{\text{th}}$ image denoted as $s_k$ with corresponding coordinate $\mathbf{X}(s_k)$) uniformly spaced in arc length.
c. Collect all the reactive single-cell trajectories that start from the epithelial region and end in the mesenchymal region.
d. For a given trial RC, divide the multi-dimensional state space by a set of Voronoi polyhedra containing individual images, and calculate the score function $F$ given in the main text (with $w = 10$ in our calculations). We carried out the minimization procedure through an iterative process. For a given trial path defined by the set of image points, we calculated a set of average points using the following equations, $\bar{\mathbf{X}}(s_k) = \dfrac{\left\langle \sum_u \sum_\alpha \left\{ \mathbf{X}_{u,t_\alpha} | \mathbf{X}_{u,t_\alpha} \in s_k \right\} \right\rangle + w \left\langle \sum_u \mathbf{X}_{u,\text{argmin}\|\mathbf{X}(s_k) - \mathbf{X}_{u,t_\alpha}\|^2} \right\rangle}{1+w}$. Next we updated the continuous reaction path through cubic spline interpolation of the average positions (*Virtanen et al., 2020*), and generated a new set of $N$ images $\{X(s_k)\}$ that are uniformly distributed along the new reaction path. We set a smooth factor, that is, the upper limit of $\sum_{k=1}^{N} (\bar{X}(s_k) - X(s_k))$, as one for calculating the RC in *Figure 4*.
e. We iterated the whole process in step three until there was no further change of Voronoi polyhedron assignments of the data points.

f. For obtaining the quasi-potential of a larger range of $s$, extrapolate the obtained reaction path forward and backward by adding additional image points (three for the two parallel reaction paths) beyond the two ends of the path linearly, respectively. These new image points are also uniformly distributed along $s$ as the old image sets do. Re-index the whole set of image points as $\{s_0, s_1, ..., s_i, ...s_N, s_{N+1}\}$.

## Calculation of dynamics of morphology and Haralick features along reaction path

The reaction path is calculated in the principal component (PC) space of morphology PC1, vimentin Haralick PC1, PC3, and PC4. Distribution of cells show significant shift before and after TGF-β treatment in these dimensions (*Wang et al., 2020*). To reconstruct dynamics in the original features space from PCs, the reaction path's coordinates on the other dimensions of PC are set as means of data in the corresponding Voronoi cell of each point on the reaction path. We obtain the reaction path in full dimension of PC space. The dynamics of morphology and Haralick features are calculated by inverse-transform of coordinates of PCs.

## Markov property test

We examined the Markov property of the RC trajectories with Chapman-Kolmogorov Test (CK-test) (*Figure 5—figure supplement 1*). The CK-test compares the left and right sides of Chapman-Kolmogorov equation ($P(k\tau) = p^k(\tau)$). The $k$-step transition matrix should equal to the $k_{th}$ power of 1-step transition matrix if the process is Markovian. We estimated the transition matrix with PyEMMA (*Scherer et al., 2015*).

## Reconstruction of quasi-potential along the reaction coordinate

Based on the theoretical framework in *Figure 5c*, we followed the procedure below:

a. The N + 2 image points of an identified RC divide the space in N + 2 Voronoi cells that data points can assign to. Ignore the first and last Voronoi cells, and use the remaining $N$ cells for the remaining analyses.

b. Within the $i_{th}$ Voronoi cell, calculate the mean drift speed (and thus $d\phi/ds$) at $s_i$ approximately by

$$\frac{d\phi}{ds}\bigg|_{s_i} = -\left\langle \frac{ds_i}{dt} \right\rangle \approx \left\langle \frac{s\left(X(t+\Delta t)\right) - s_i}{\Delta t} \right\rangle\bigg|_{s(X(t))=s_i}$$ where $s(\mathbf{X})$ is the assumed value along $s$ for a cell state

$\mathbf{X}$ in the morphology/texture feature space. The sum $s(X(t)) = s_i$ is over all time and all data points from all the recorded trajectories that lie within the $i_{th}$ Voronoi cell ($s(X(t)) = s_i$), and $\Delta t = 1$ is one recording time interval. Using data points from all instead of just reactive trajectories is necessary for unbiased sampling within each Voronoi cell with $\langle \eta \rangle_{s_i} = 0$.

c. Calculate the quasi-potential through numerical integration, $\phi(s_i) = \phi(s_0) + \int_{s_0}^{s_i} \frac{d\phi}{ds} ds \approx \phi(s_0) + \Delta s \sum_{j=1}^{i} \frac{d\phi}{ds}|_{s_j}$. The exact value of $\phi(s_0)$ does not affect the quasi-potential shape.

## Reconstruction of quasi-potential along parallel paths

We followed the same mapping procedure for trajectories of cells treated with different TGF-β concentration. We used *tslearn* to calculate the DTW distance between two reactive trajectories, then performed K-Means clustering (*Sammut and Webb, 2017*) on the DTW distance matrix (*Sakoe and Chiba, 1978*) to cluster the reactive trajectories into two groups. We then followed the procedure in Materials and methods (*McFaline-Figueroa et al., 2019*) to reconstruct the RC for each group. We reconstructed the quasi-potentials using all trajectories.

For a single-cell trajectory, it is possible that the cell jumps out its original path due to fluctuation. For example, for a trajectory that mainly follows RC1, certain parts of it may transit into the range of RC2. So we used a part-aligning method to map all the trajectories to the RCs.

For a trajectory not belonging to the reactive trajectory ensemble, we assigned it to one of the two group associated to the two RCs, $\{s_1\} = \{s_{1,1}, ..., s_{1,i}, ...s_{1,N}\}$ and $\{s_2\} = \{s_{2,1}, ..., s_{2,i}, ...s_{2,N}\}$ by using sub-sequence DTW distance (*Sammut and Webb, 2017*). We first calculate the sub-sequence DTW

distance of this trajectory to the Voronoi cells of $\{S_1\}$ and $\{S_2\}$, respectively, and identified its matching coordinates $\{s_{1,a}, ..., s_{1,i}, ..., s_{1,b}\}$, and $\{s_{2,c}, ..., s_{2,i}, ...s_{2,d}\}$ on the RCs. Each point on the trajectory was assigned to the RCs based on minimum Euclidean distance. Then the consecutive parts of this trajectory along each RC were used to calculate the drift speed $\langle ds/dt \rangle$ and quasi-potential $\phi(s)$ following the definition and procedure described in Materials and methods (*McFaline-Figueroa et al., 2019*).

## Numerical solution of the Fokker-Planck equation

The Langevin equation, $ds/dt = F(s) + \eta(s, t)$, describes how one cell evolves along a reaction coordinate $s$ in the state space. Equivalently under the Ito interpretation the Fokker-Planck equation describes on the probability density of observing cells at a specific point $s$, $\rho(s, t)$ changes over time (*Kubo et al., 1991*; *Schnoerr et al., 2017*),

$$\partial_t \rho(s, t|s_0, t_0) = \nabla \cdot \left[ D \left( \nabla - \frac{F(s)}{k_B T} \right) \rho(s, t|s_0, t_0) \right].$$

In the equation, the spatially dependent diffusion constant D was estimate from the experiment data, $D_{s_i} = Var\left(\frac{ds}{dt}|_{s_i}\right)$, the variance of $\frac{ds}{dt}|_{s_i}$ calculated from experiment data in corresponding Voronoi cells. The term $F(s) = -\nabla U$, where $U$ is the quasi-potential obtained from experiment data, the pseudo-temperature $T$ was set as $\frac{1}{2k_B}$, where $k_B$ is the Boltzmann constant. With these inputs, we solved the equations numerically (*Holubec et al., 2019*).

## Data availability

The scripts and single-cell trajectory data can be found in the link https://github.com/opnumten/trajectory_analysis (*Wang, 2022* copy archived at swh:1:rev:1153076f88acab6f1030eefa119f4e6bd49cbb97).

## Acknowledgements

This work was partially supported by National Cancer Institute (R37 CA232209), National Institute of Diabetes and Digestive and Kidney Diseases (R01DK119232) to JX, and National Institutes of Health (NIBIB T32EB009403) to DP. This work used the Extreme Science and Engineering Discovery Environment (XSEDE) at SDSC Dell Cluster with NVIDIA V100 GPUs NVLINK and HDR IB (Expanse GPU) through allocation BIO200085. We thank Eric Siggia, John J Tyson, and Sophia Hu for critical reading of the manuscript and constructive comments.

## Additional information

### Funding

| Funder | Grant reference number | Author |
| --- | --- | --- |
| National Institute of Diabetes and Digestive and Kidney Diseases | R01DK119232 | Jianhua Xing |
| National Cancer Institute | R37 CA232209 | Jianhua Xing |
| National Institutes of Health | NIBIB T32EB009403 | Dante Poe |

The funders had no role in study design, data collection and interpretation, or the decision to submit the work for publication.

### Author contributions

Weikang Wang, Data curation, Formal analysis, Investigation, Methodology, Software, Validation, Visualization, Writing – original draft, Writing – review and editing; Dante Poe, Yaxuan Yang, Thomas Hyatt, Investigation, Writing – review and editing; Jianhua Xing, Conceptualization, Formal analysis, Funding acquisition, Investigation, Methodology, Project administration, Resources, Supervision, Visualization, Writing – original draft, Writing – review and editing

## Author ORCIDs

Weikang Wang (iD) http://orcid.org/0000-0002-4783-7540
Dante Poe (iD) http://orcid.org/0000-0002-4813-3035
Yaxuan Yang (iD) http://orcid.org/0000-0003-3077-2813
Thomas Hyatt (iD) http://orcid.org/0000-0001-7536-0850
Jianhua Xing (iD) http://orcid.org/0000-0002-3700-8765

## Decision letter and Author response

Decision letter https://doi.org/10.7554/eLife.74866.sa1
Author response https://doi.org/10.7554/eLife.74866.sa2

---

# Additional files

## Supplementary files
• Transparent reporting form

## Data availability

The computer code are shared on GitHub, so other researchers can run to reproduce Figure 3, 4, and 5. The processed single cell trajectory data are on Dryad.

The following dataset was generated:

| Author(s) | Year | Dataset title | Dataset URL | Database and Identifier |
| --- | --- | --- | --- | --- |
| Wang W | 2022 | A549 VIM-RFP EMT single cell trajectory data | https://dx.doi.org/10.5061/dryad.7h44j0zvp | Dryad Digital Repository, 10.5061/dryad.7h44j0zvp |

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
