## [Editor Report]

This is a multifaceted study of the epithelial to mesenchymal transition (EMT) in live cells. EMT is relevant for cancer, development, and wound healing. The authors were able to discern two possible cell transition path categories without multi-color labeling or other advanced experimental approaches, which could be impactful for other studies. The study draws on a wide range of experimental, data science, and modelling tools and techniques.

---

## [Decision Letter]

**Decision letter after peer review:**

Thank you for submitting your article "Epithelial-to-mesenchymal transition proceeds through directional destabilization of multidimensional attractor" for consideration by *eLife*. Your article has been reviewed by 3 peer reviewers, and the evaluation has been overseen by a Reviewing Editor and Aleksandra Walczak as the Senior Editor. The following individuals involved in review of your submission have agreed to reveal their identity: Gabor Balazsi (Reviewer #1); Michael Stumpf (Reviewer #2); Jian Liu (Reviewer #3).

Overall, all three reviewers are positive about your work. However, there are items that need to be addressed, which are summarised below. Note that the three reviews are attached in their entirety to assist your revision.

Essential points to be addressed:

1. How the existence of multiple reaction paths is or is not robust to e.g. multiplicative noise and other factors in the analysis.

2. What is the method's applicability to other cell lines, proteins, and cellular transitions.

3. Whether a simpler analysis would have given the same results.

4. Clearly state what the advance this paper has over the previous Science Advance paper (Ref. 11).

5. Improve clarity of your writing, including tidying up typos and references, explaining to biology audience the difference between Langevin and Fokker-Planck equations, providing justification of using Langevin, discussing how this system's transition is different from the typical analysis, explaining the plethora of computational methods deployed, and providing better illustration of morphological and textural features.

*Reviewer #1 (Recommendations for the authors):*

(1) In the Introduction, the definition of the saddle-node bifurcation is unusual. Specifically, the saddle-node bifurcation is described here as both the appearance of a saddle point and a new steady state, and then the merger of the saddle point with the original steady state. Typically, however, these are considered two different saddle-node bifurcations. In fact, there are systems exhibiting a single saddle-node bifurcation, when the saddle point never merges with the original steady state, so the system remains bistable even as the bifurcation parameter tends to infinity. See the wild-type circuit in PMID: 31754027 for an example.

(2) The way the system's transition is induced here is different from the way typical bifurcations are realized mathematically. Specifically, in mathematics the system's steady states are investigated while a bifurcation parameter is scanned – but each particular value of the bifurcation parameter is considered fixed. In contrast, in this experimental system the bifurcation parameter is time-dependent: intracellular TGF-β concentrations change as they equilibrate with the extracellular TGF-β levels. This difference should be mentioned and discussed, regarding the influx rate of TGF-β and membrane permeability. About how long does it take for intracellular and extracellular TGF-β to equilibrate?

(3) Related to the previous point, an experiment closer to the mathematical studies would keep cells in microfluidic chambers, perfusing media with constant TGF-β concentration that does not depend on time but may increase from chamber to chamber. This should be mentioned as a possibility for future studies.

(4) The analysis involves a plethora of computational and mathematical methods: PCA, active shape model, Haralick features, SOM, Dijkstra algorithm, Onsager-Machlup action, dynamics time warping, Focker-Planck equation, pseudopotential, Voronoi partitioning, finite temperature string method, etc. While all of this is impressive, there is a concern. Some parts of the text are difficult to follow, and the overall procedure is not easy to comprehend. To make this approach really useful to the EMT community, a knowledgeable biologist should be able to comprehend and reproduce the results. So, is it possible to get to the same conclusion with fewer and/or simpler computational steps? Are all the steps employed here necessary to uncover the two cell transition path categories? Moreover, how transferable is the methodology to other cell lines, other CPTs and other proteins? A highly sophisticated, multistep approach is probably less likely to transfer and generalize than a simpler one. All of this should be at least discussed, and possibly addressed by an attempt to simplify the steps, accordingly simplifying the text of the manuscript by reducing some of the jargon.

(5) Most of the methods are introduced and similar conclusions about the CPT paths are reached in the earlier Science Advances paper (Ref. 11.) using the same cell line, the same markers, and the same CPT. The authors should clearly state what is distinguishingly novel in this manuscript compared to their earlier publication.

(6) A more intuitive illustration of the morphology and texture features is needed. What cellular aspects do the main Principal Components contain? Are any morphology and texture aspects overrepresented in the first PCs? For example, it would be very helpful to illustrate the morphology and texture extracted, along with their computational analysis to obtain the numbers for a few cells: (i) in the E state; (ii) in the M state; (iii) in mid-transition with vimentin increases first; (iv) in mid-transition with concordant increase. This could be addressed by altering the current Figure 4, similar to Figures 3 and 4 in the Science Advances paper (Ref. 11.).

(7) The Langevin approach assumes uncorrelated Gaussian noise. However, most fluctuations of cellular molecules are neither Gaussian, nor uncorrelated. Moreover, the noise properties can depend on the deterministic components of the Langevin approach: F, x and t. What justifies the applicability of the approach? This should be discussed.

(8) For the 1 ng/mL TGF-β treatment, what would the actual reaction path look like (compared to the 4 ng/mL treatment)? Here the new trajectories are projected onto the RCs for the 4 ng/mL treatment "for comparison". What does this projection imply compared to a separate analysis of the 1 ng/mL treatment? What if the reverse is done: the trajectories of the 4 ng/mL treatment are projected onto the RCs of the 1 ng/mL treatment?

(9) Related to the above question, it is interesting that the new M attractor for 1 ng/mL treatment is closer to the E state. Does this mean that the attractor-based definition of the "M" state is stimulus-dependent? So how can we be sure there is an "E" and an "M" state if their definition changes according to the environment? Shouldn't these states be independent of the environment?

(10) It is interesting that some cells become mesenchymal in low TGF-β, and possibly even without TGF-β. Do any of these cells revert? It is understood that high TGF-β prevents reversion, but the plateau in Figure 5 implies that reversions may be possible at low or zero TGF-β.

(11) Are there any "early signs" of cells that will become mesenchymal? Could this be predicted based on the values, or fluctuations of numerical features or their correlation analysis?

(12) This manuscript is about environment-induced CPTs. On the other hand, CPTs can also occur stochastically in a constant environment. This distinction would be worth making to place the research in context, citing PMID:21414483.

(13) A reference is needed for TPT in line 151 and also for the Onsager-Machlup action in line 200.

(14) The phrase "round polygon" probably means convex polygon?

(15) CPT appears in the Abstract without a prior definition.

(16) References (16) and (17) are identical.

(17) Please watch the grammar! There are some missing or extra words, e.g., "instant OF time", "towards TO", etc.

*Reviewer #2 (Recommendations for the authors):*

Specific points:

Figure 1: The figure legend could be made clearer. The figure might give the impression that the data is sufficient to distinguish between saddle node and pitchfork bifurcation.

line 135: I would like to see an outline of what Haralick features are.

line 199-200: It would help to define in general terms what an action is.

line 203: reference 22 has nothing to do with reaction coordinates as far as I can make out. Did the bibliography get mixed up here?

line 262: in 1D all dynamical systems are gradient systems. References 28 and 29 are not the most appropriate references in this context. Most introductory dynamical systems books would suffice.

line 283-284: It may help some readers to learn more about the differences between Langevin and Fokker-Planck equations. Schnoerr et al., Journal of Physics A (2017) 50:093001 is a reference that I find very useful.

line 287: Is it possible to do better than "matches reasonably well"? Are there statistical measures by which this can be quantified? Or is it possible to explain why there is a mismatch.

line 319-323: I found the discussion about the intermediate states fascinating: I was wondering if this could be extended to include some of the arguments of PMC6238957 or similar? More generally, mathematically, for the systems considered here (in a deterministic regime) the Palais-Smale conditions or the mountain pass theorem would hold. MacLean et al., make such arguments less formally and more intuitive.

Finally, most other previous authors appear to have used the term quasi-potential to denote landscapes of differentiating systems. In solid-state theory and chemistry pseudo-potential appears to be favoured to describe e.g. effective electron potentials. I would recommend the terminology "quasi-potential" here.

*Reviewer #3 (Recommendations for the authors):*

1. The writing needs to be greatly improved. While some parts are arguably a subject of style/taste, the rest of the manuscript is littered with grammar mistakes. For instance, the "CPT" in Abstract needs to be defined first. On Lines 37-38: "different function, morphology, …" should be "different functions, morphologies…". On Line 48-49: "A cell is a dynamical system, and understanding a CPT process from dynamical systems theory …" should be something like "Considering cell as a dynamical system, understanding a CPT process from dynamical systems theory…".

2. Any results from deep learning critically hinge on the quality of the training set; otherwise, the automation can easily go wrong. In automatically characterizing the live-cell time-lapsed images, the authors need to provide the necessary baseline or the control in their deep learning method. If it is already done in their previous work, then the authors need to explicitly state and refer to it in the current paper. If not, then such a control measure in deep learning needs to be included in the Method.

3. Using time-lapsed images to reconstruct pseudopotential is a great improvement over the previous work. The question: How does the number of images points or the time-resolution along the reaction coordinate affect the reconstructed potential? The authors need to at least discuss the potential effects.

4. With the high-dimensional parameter space, the authors reconstructed the common transition paths of EMT. It is well known that cells exhibit large heterogeneity in terms of gene expression and dynamics. The question is: How can we reconcile the two opposing features?

5. The authors demonstrated the two parallel pathways in EMT with the same starting and ending states (e.g., Figure 3e). While the reaction coordinates of transition state along one pathway (vimentin PC1 and morphology PC1) are intermediate between the E and the M states (the right panel of Figure 3e), those along the other pathway are not (the left panel of Figure 3e). What is the physical nature of this largely non-monotonic change? And if possible, what is the functional role? In perspective, cell operates in the multi-dimensional parameter space. What the authors have characterized is only the subset. Possibly, there exist additional but essential parameters that remain to be explored. This way, the non-monotonic change in the reaction coordinates may reflect the projection from a higher dimensional space onto the two-dimensional parameter plane. For instance, cell mechanics may be another set of key parameters that underlie EMT, which has been demonstrated to display non-monotonic changes during EMT (see Margaron et al., Biophysical properties of intermediate states of EMT outperform both epithelial and mesenchymal states (bioRxiv, 2019)). I'd suggest the authors discuss the finding in a broader context.

---

## [Author Response]

Essential points to be addressed:1. How the existence of multiple reaction paths is or is not robust to e.g. multiplicative noise and other factors in the analysis.

The existence of the two paths was revealed from clustering the trajectories. We obtained the two paths with several different clustering approaches. All these analyses do not require any assumption on the underlying dynamics.

2. What is the method's applicability to other cell lines, proteins, and cellular transitions.

This method is general for studying cell phenotypic transitions, and can be used on other cell lines and cellular processes. In a separate work (Wang et al., BioRxiv, 2021.09.21.461257, doi: https://doi.org/10.1101/2021.09.21.461257), we also applied the transition path analyses on scRNA-seq data.

3. Whether a simpler analysis would have given the same results.

This work has two parts.

In part 1, we analyzed the single cell trajectories using some standard clustering algorithms (used in different contexts), and the analyses revealed two parallel transition paths. As discussed in point 1, the analyses are standard in data sciences.

In part 2, we applied the transition path analyses to the single cell trajectories, and this part also has two steps:

1) We adapted the finite temperature string method widely used in studying chemical reactions to reconstruct the reaction coordinates for the two paths. Notice up to this step (i.e., part 1 and step 1 of part2) we had not made any assumption on the system dynamics. Here we mentioned Onsager-Machlup action only to give a qualitative theoretical explanation on why cells form concentrated reaction tubes in the state space. In this revised version we removed this part to avoid reference to unnecessary theoretical discussions.

2) We reconstructed the equations of motion along the reaction coordinates under a minimal ansatz of Langevin dynamics.

Indeed some of the theoretical concepts and approaches are from chemical physics, and may not be familiar to the quantitative cell biology field. Introduction of these approaches is an important and exciting aspect of this work to demonstrate that these theoretical and computational tools conventionally used in chemical physics can also be applied to analyze single cell live cell imaging data; compared to typical comprehensive and often very expensive experimental measurements such as single cell genomics studies, we show how these tools can reveal quantitative mechanistic information from the often-regarded-as-less-informative imaging data. That is, our integrated live cell imaging and analysis platform provides dynamical information complements and corroborates with what one can get from snapshot single cell data, which can only provide partial dynamical information due to the destructive nature of the techniques used.

4. Clearly state what the advance this paper has over the previous Science Advance paper (Ref. 11).

We added in paragraph 1 of the Discussion section. In the Science Advance paper, we established a live cell imaging and image analysis procedure so one can represent cell phenotypic transitions mathematically as trajectories in a composite cell feature space, analogous to how one represents molecular motions in a phase space. In this work, with the mathematical representation of live cell imaging data we applied concepts and techniques developed in chemical physics for studying molecular processes to investigate cell phenotypic transitions. With the quantitative approaches we reconstructed the reaction coordinates and the equations of motion along the transition paths, and the quantitative results suggest a sequential saddle-node bifurcation mechanism for TGF-β induced EMT in A549 cells.

5. Improve clarity of your writing, including tidying up typos and references, explaining to biology audience the difference between Langevin and Fokker-Planck equations, providing justification of using Langevin, discussing how this system's transition is different from the typical analysis, explaining the plethora of computational methods deployed, and providing better illustration of morphological and textural features.

Thanks for the suggestions. We have made corresponding changes.

Reviewer #1 (Recommendations for the authors):(1) In the Introduction, the definition of the saddle-node bifurcation is unusual. Specifically, the saddle-node bifurcation is described here as both the appearance of a saddle point and a new steady state, and then the merger of the saddle point with the original steady state. Typically, however, these are considered two different saddle-node bifurcations. In fact, there are systems exhibiting a single saddle-node bifurcation, when the saddle point never merges with the original steady state, so the system remains bistable even as the bifurcation parameter tends to infinity. See the wild-type circuit in PMID: 31754027 for an example.

Thanks for pointing out. Indeed our previous wording is not precise. We have modified the discussions.

(2) The way the system's transition is induced here is different from the way typical bifurcations are realized mathematically. Specifically, in mathematics the system's steady states are investigated while a bifurcation parameter is scanned – but each particular value of the bifurcation parameter is considered fixed. In contrast, in this experimental system the bifurcation parameter is time-dependent: intracellular TGF-β concentrations change as they equilibrate with the extracellular TGF-β levels. This difference should be mentioned and discussed, regarding the influx rate of TGF-β and membrane permeability. About how long does it take for intracellular and extracellular TGF-β to equilibrate?

This is a good question, and we added discussions in the paragraph (lines 406-411). The bifurcation parameter here is the extracellular concentration of the exogenous TGF-β, as in our previous studies and typical bifurcation analyses. As the reviewer pointed out, intracellular TGF-β levels are more complex and difficult to use as a bifurcation parameter. The transduction of TGF-β requires time. The time scale of dynamics of tyrosine kinase receptor is in minutes level (1). But the downstream like Smad phosphorylation and translocation to nucleus need several hours(2, 3). Since the exogenous TGF-β is externally controlled and is in large excess, to a good approximation we treated its level as a constant.

(3) Related to the previous point, an experiment closer to the mathematical studies would keep cells in microfluidic chambers, perfusing media with constant TGF-β concentration that does not depend on time but may increase from chamber to chamber. This should be mentioned as a possibility for future studies.

Thanks. We added discussions in the paragraph (lines 406-411). One should be aware of one complexity though. Cells also secrete endogenous TGF-β. So such a microfluidic experiment corresponds to the condition of a constant total TGF-β concentration.

(4) The analysis involves a plethora of computational and mathematical methods: PCA, active shape model, Haralick features, SOM, Dijkstra algorithm, Onsager-Machlup action, dynamics time warping, Focker-Planck equation, pseudopotential, Voronoi partitioning, finite temperature string method, etc. While all of this is impressive, there is a concern. Some parts of the text are difficult to follow, and the overall procedure is not easy to comprehend. To make this approach really useful to the EMT community, a knowledgeable biologist should be able to comprehend and reproduce the results. So, is it possible to get to the same conclusion with fewer and/or simpler computational steps? Are all the steps employed here necessary to uncover the two cell transition path categories? Moreover, how transferable is the methodology to other cell lines, other CPTs and other proteins? A highly sophisticated, multistep approach is probably less likely to transfer and generalize than a simpler one. All of this should be at least discussed, and possibly addressed by an attempt to simplify the steps, accordingly simplifying the text of the manuscript by reducing some of the jargon.

From the perspective of methodology, active shape model and Haralick features are what we use to represent the single cell dynamics mathematically. We believe that there are other ways to represent the single cell dynamics. For example, researchers can use other morphology features like area, major axis length and perimeter. But as mentioned in a systematic study of Pincus et al(4)., the active shape mode with PCA is one of the best representations of cell morphology. While active shape model is abstract, the PCA can reveal the general feature of variation In active shape model. For instance, our analysis show that the first principal component mode in EMT of A549 treated with TGF-β is close to the variation of major axis length. Haralick features are one type of texture features. Researchers can use other types of texture features like wavelet features, moment features (5). For this step, one just needs find the representation that is suitable for their data.

The active shape mode and Haralick features can be used on other types of cell phenotype transition. We also have applied the analysis pipeline including the Dijkstra algorithm, finite temperature string method and Voronoi partitioning on single cell RNA sequencing data (https://www.biorxiv.org/content/biorxiv/early/2021/09/24/2021.09.21.461257.full.pdf).

Dynamic time warping has been commonly used method in time series analysis. SOM and Dijkstra algorithm are alternative to the dynamic time warping method. We recovered two parallel EMT paths using these two different analysis approaches. The finite temperature string method, Voronoi partitioning and reconstruction of pseudopotential are chemical physics approaches typically in studying chemical reactions.

An important part of the present work is to show applying these analysis tools originally developed in various fields, one can extract quantitative mechanistic information from simple bright field/fluorescent images, as compared the much more complex and expensive single cell genomics approaches. We summarize the analysis pipeline using one schematic single cell trajectory (Figure 5—figure supplement 7). The code will be publicly available on Github.

(5) Most of the methods are introduced and similar conclusions about the CPT paths are reached in the earlier Science Advances paper (Ref. 11.) using the same cell line, the same markers, and the same CPT. The authors should clearly state what is distinguishingly novel in this manuscript compared to their earlier publication.

We added discussions in the paragraph (lines 348-357). In the Sci Adv paper we established a framework of recording and representing mathematically single cell trajectories in multi-dimensional composite feature space. In this work we developed a downstream theoretical framework of analyzing the recorded single cell trajectories. Through experimenting on A549 cells without any treatment and A549 cells treated with 1, 4 ng/ml TGF-β, we revealed a sequential saddle-node bifurcation mechanism.

(6) A more intuitive illustration of the morphology and texture features is needed. What cellular aspects do the main Principal Components contain? Are any morphology and texture aspects overrepresented in the first PCs? For example, it would be very helpful to illustrate the morphology and texture extracted, along with their computational analysis to obtain the numbers for a few cells: (i) in the E state; (ii) in the M state; (iii) in mid-transition with vimentin increases first; (iv) in mid-transition with concordant increase. This could be addressed by altering the current Figure 4, similar to Figures 3 and 4 in the Science Advances paper (Ref. 11.).

Thanks for the advice and we modified Figure 4 accordingly. We added representative single cell images of different types (E state, M state, in mid-transition with vimentin increases first and in mid-transition with concordant increase).

For the morphology space, the first principal component of morphology mainly reflects the variation of major axis length. And the second principal component reflects the variation in the minor axis length. Both PCs are related to cell area.

For Haralick features, the coefficients of different features in the first component are quite similar. In the third principal component, correlation and information measure of correlation 2 count for large proportion. In the fourth principal component, entropy and information measure of correlation 1 play important roles.

(7) The Langevin approach assumes uncorrelated Gaussian noise. However, most fluctuations of cellular molecules are neither Gaussian, nor uncorrelated. Moreover, the noise properties can depend on the deterministic components of the Langevin approach: F, x and t. What justifies the applicability of the approach? This should be discussed.

Thanks. The Langevin ansatz used in this work assumes multiplicative noises in general. We made it clear by explicitly adding the x-dependence in the equation. See also description in the method section 9. Notice we performed CK tests to validate the Markovian property. Given the noisy data and a limited number of trajectories we could obtain with the current experimental setup, in this study we restrict our analysis to this minimal model, and leave more complex models to future studies.

(8) For the 1 ng/mL TGF-β treatment, what would the actual reaction path look like (compared to the 4 ng/mL treatment)? Here the new trajectories are projected onto the RCs for the 4 ng/mL treatment "for comparison". What does this projection imply compared to a separate analysis of the 1 ng/mL treatment? What if the reverse is done: the trajectories of the 4 ng/mL treatment are projected onto the RCs of the 1 ng/mL treatment?

We calculated the RCs for the 1 ng/ml case (Author response image 1). Compare with the 4ng/ml treatment, the final state of 1 ng/ml treatment is closer to the initial epithelial state. The corresponding quasi-potentials are qualitatively similar to what shown in Figure 5d. We decide not to include in the manuscript given that there is already lot of information.

**Author response image 1. sa2fig1:** Transition path analyses of single cell trajectories along the RCs obtained from 1 and 4 ng/ml trajectories, respectively. (a) RCs in the 2-D state space plane. (b) Quasi-potentials of 1 ng/ml trajectories reconstructed along the 1 ng/ml RCs. Left: Vimentin varies first. Right: Concerted variation.

The reason we projected the 1 ng/ml trajectory data is to compare the two different situations on the same coordinates. The reverse analysis is less meaningful because that a large part of the data points in 4 ng/ml treatment will be projected to the final coordinate of 1 ng/ml. This will give a result that cells in 4 ng/ml will locate in the final state of 1 ng/ml treatment, which is not the case.

(9) Related to the above question, it is interesting that the new M attractor for 1 ng/mL treatment is closer to the E state. Does this mean that the attractor-based definition of the “M” state is stimulus-dependent? So how can we be sure there is an“E” and an“M" state if their definition changes according to the environment? Shouldn't these states be independent of the environment?

This is really a good question. Yes, the attractor-based definition is stimulus-dependent. It is apparent from a mathematics perspective, for example, the state on a branch of a bifurcation diagram generally changes with the bifurcation parameter. This dependence has been also gradually accepted in the EMT field, and researchers use quantities like EMT score to describe the continuum spectrum of EMT. As suggested in (6), one may use some cellular features and properties to define different EMT phenotypes. In this work we showed that cells with and without TGF-β occupy distinct regions in the vimentin texture feature – morphological feature space. In our Science Advance paper, we also showed corresponding changes of selected EMT markers and transcription factors.

(10) It is interesting that some cells become mesenchymal in low TGF-β, and possibly even without TGF-β. Do any of these cells revert? It is understood that high TGF-β prevents reversion, but the plateau in Figure 5 implies that reversions may be possible at low or zero TGF-β.

It is a question we would like to explore. From our recorded single cell trajectories, some of the cells jumping out of the epithelial region will transit back into the epithelial region we defined in the composite feature spaces. Author response image 2 shows one representative 1 ng/ml trajectory. Several studies in other systems suggest such dynamic equilibrium among different sub-phenotypes(7-9), and it is a problem our integrated live cell imaging and image analysis platform can provide quantitative information.

**Author response image 2. sa2fig2:** A representative reactive trajectory which transits back into the epithelial region. The arrow indicates the end point.

(11) Are there any“"early sign”" of cells that will become mesenchymal? Could this be predicted based on the values, or fluctuations of numerical features or their correlation analysis?

Thanks for the advice. This is a question that we are currently investigating. We compared the initial distribution of cells transiting into mesenchymal state and that of cells failing to transit into mesenchymal state. We found no difference between distributions of their initial conditions (Author response image 3).

**Author response image 3. sa2fig3:** Distribution of initial values of EMT trajectory and failed EMT trajectory. a: Distribution of initial morphology PC1; b: Distribution of initial vimentin Haralick PC1.

For bulk data, there are studies on identifying the early warning signals of critical transitions. We utilized the critical index of Mojtahedi et al., (10). This critical index is defined as Ic=⟨│R(fi,fj)│⟩⟨R(Sk,Sl)⟩ , where f are the feature (or gene) vectors, are the cell state vectors at sampling time t, and R are the Pearson’s correlation coefficients. As shown in Author response image 4, we can identify the time when transition in population level happens (peak). We also modified it to apply on the single trajectories with a sliding window method. But the limitation is that frequency of the experiment data is not high enough to support such analysis, so we are working on improving it.

Most of existing methods on early warning signs can only be used on 1-D time series (11). We are working on finding a method to identify the early sign of critical transition in multi-dimensional trajectories. We also hypothesize that our cell state resolution may not be sufficient to distinguish the two populations. Currently we are using a cell line with additional cell cycle reporter and microscopes that can provide more cell feature information.

**Author response image 4. sa2fig4:** Critical state index along time. The critical state index shows a peak around 24 hour, which indicates the transition from epithelial state to mesenchymal state.

(12) This manuscript is about environment-induced CPTs. On the other hand, CPTs can also occur stochastically in a constant environment. This distinction would be worth making to place the research in context, citing PMID:21414483.

Thanks for pointing this out. We added the reference in our discussion.

(13) A reference is needed for TPT in line 151 and also for the Onsager-Machlup action in line 200.

Thanks. We add TPT references in the manuscript, and removed reference to the Onsager-Machlup action.

(14) The phrase "round polygon" probably means convex polygon?

Yes. Thanks for pointing this out. We modified it.

(15) CPT appears in the Abstract without a prior definition.

Thanks. We modified it.

(16) References (16) and (17) are identical.

Thanks. We modified it.

(17) Please watch the grammar! There are some missing or extra words, e.g., "instant OF time", "towards TO", etc.

Thanks. We modified the manuscript.

Reviewer #2 (Recommendations for the authors):Specific points:Figure 1: The figure legend could be made clearer. The figure might give the impression that the data is sufficient to distinguish between saddle node and pitchfork bifurcation.

Thanks. We modified the figure caption.

line 135: I would like to see an outline of what Haralick features are.

We provided the definition of Haralick features in the caption of Figure 4.

We described the calculation of Haralick features in Materials and methods (3). The calculation is based on the gray-level co-occurrence matrix G. Gij is the frequency when two neighbor pixels’ values are i and j. It is normalized by the total counts. In 2D images, there are four directions of such neighbors *(0°, 45°, 90°, and 135°).* To be rotation invariant, each feature is calculated on all the four directions of G matrix and averaged over them. Haralick features include the following features: 1 Angular Second Moment; 2: Contrast; 3: Correlation; 4: Sum of Squares: Variance; 5: Inverse Difference Moment; 6: Sum Average; 7: Sum Variance; 8: Sum Entropy; 9: Entropy; 10: Difference Variance; 11: Difference Entropy; 12: Information Measure of Correlation 1; 13: Information Measure of Correlation 2.

line 199-200: It would help to define in general terms what an action is.

We deleted discussions about actions since the concept was mentioned in the previous version only for understanding the transition tubes.

line 203: reference 22 has nothing to do with reaction coordinates as far as I can make out. Did the bibliography get mixed up here?

Thanks for point this out. We modified this reference.

line 262: in 1D all dynamical systems are gradient systems. References 28 and 29 are not the most appropriate references in this context. Most introductory dynamical systems books would suffice.

Thanks. We modified it and added more references.

line 283-284: It may help some readers to learn more about the differences between Langevin and Fokker-Planck equations. Schnoerr et al., Journal of Physics A (2017) 50:093001 is a reference that I find very useful.

Thanks. We added a brief introduction of both equations in Materials and methods (11).

line 287: Is it possible to do better than "matches reasonably well"? Are there statistical measures by which this can be quantified? Or is it possible to explain why there is a mismatch.

We compared the predicted steady distributions with the experiment data distributions of both paths. For the path that vimentin varies first, the mean value and standard deviation of the predicted steady distribution are 17.2 and 4.6. And the mean value and standard deviation of distribution of experiment data are 16.0 and 5.0. For the path of concerted variation, the mean value and standard deviation of the predicted steady distribution are 14.7 and 4.4. And the mean value and standard deviation of distribution of experiment data are 16.2 and 4.0. From the perspective of data analysis, the discretization of state space is one source of this error. If we use a smaller number of reaction coordinate, the mean squared error will increase. A more coarse-grained Voronoi separation increase this error when discretizing the trajectory progression. More importantly, our analysis is based on two 1-D paths analysis instead of direct 2D analysis. We assign the data points by their dynamics (dynamic time warping) to their corresponding path. But the probable wrong assignment would affect the calculation of the potential.

line 319-323: I found the discussion about the intermediate states fascinating: I was wondering if this could be extended to include some of the arguments of PMC6238957 or similar? More generally, mathematically, for the systems considered here (in a deterministic regime) the Palais-Smale conditions or the mountain pass theorem would hold. MacLean et al., make such arguments less formally and more intuitive.

Thanks. We added some discussion on the intermediate states in the manuscript. The papers you raised are highly relevant. We are in the process of reconstructing the multi-dimensional vector field, and from examining these vector fields we may get more insights on addressing the question. In a separate study (12), we reconstructed the vector field in the expression space, and were able to perform topological characterizations.

Finally, most other previous authors appear to have used the term quasi-potential to denote landscapes of differentiating systems. In solid-state theory and chemistry pseudo-potential appears to be favoured to describe e.g. effective electron potentials. I would recommend the terminology "quasi-potential" here.

Thanks. We modified all the pseudo-potential to quasi-potential.

Reviewer #3 (Recommendations for the authors):1. The writing needs to be greatly improved. While some parts are arguably a subject of style/taste, the rest of the manuscript is littered with grammar mistakes. For instance, the "CPT" in Abstract needs to be defined first. On Lines 37-38: "different function, morphology, …" should be "different functions, morphologies…". On Line 48-49: "A cell is a dynamical system, and understanding a CPT process from dynamical systems theory …" should be something like "Considering cell as a dynamical system, understanding a CPT process from dynamical systems theory…".

Thanks. We modified these errors and the writing of the manuscript.

2. Any results from deep learning critically hinge on the quality of the training set; otherwise, the automation can easily go wrong. In automatically characterizing the live-cell time-lapsed images, the authors need to provide the necessary baseline or the control in their deep learning method. If it is already done in their previous work, then the authors need to explicitly state and refer to it in the current paper. If not, then such a control measure in deep learning needs to be included in the Method.

Thanks for pointing this out. We also realized this when performing tracking on live cells. To reduce the influence of false segmentation on the trajectory analysis, we used a method to filter out those probable over- and under-segmentation. We calculated the overlap between cells in consecutive frames. The cells that have abnormal overlap values were detected. And their trajectories were analyzed. We filtered out those over- and under-segmented cells and relinked the broken trajectories with linear assignment algorithm. We added this part in Material and Methods.

3. Using time-lapsed images to reconstruct pseudopotential is a great improvement over the previous work. The question: How does the number of images points or the time-resolution along the reaction coordinate affect the reconstructed potential? The authors need to at least discuss the potential effects.

In a wide range of image point numbers, the reconstructed potentials of 4ng/ml and 1ng/ml keep the same (see new Figure 5—figure supplement 6). While more image points lead to smoother reconstructed quasi-potential, the number is compromised by keeping the number of sampling points in each Voronoi cell sufficient.

4. With the high-dimensional parameter space, the authors reconstructed the common transition paths of EMT. It is well known that cells exhibit large heterogeneity in terms of gene expression and dynamics. The question is: How can we reconcile the two opposing features?

The reviewer asked a very deep and fundamental question in studying complex systems, which have the characteristics of many coupled degrees of freedom with no clear time scale separation, and sometime not at thermodynamic equilibrium (e.g., for cells). Another example of such heterogeneity, has been also noticed in studying reactions of complex molecular systems such as enzymatic reactions (e.g., the single molecule enzymology studies of Sunney Xie), and is termed dynamic disorder in the chemical physics field (13). Existence of heterogeneity indicates existence of one or more other slow degrees of freedom that couples to the transition process under study. Therefore it is still an active research area whether and how one can use one or a few 1D transition paths to represent a transition process:

a) Different cells (or molecules in chemical reactions) share common transition mechanisms. Intuitively, for EMT while cells show large heterogeneity in their cell sizes, etc, it is straightforward to tell whether individual cells undergo EMT. One can identify common characteristic changes in terms of cell morphological features (enlarged and elongated shape), and vimentin texture features (increased expression with a change from localized to dispersive distribution).

b) Proper choice of the coordinates is essential. For a given transition process, it is known in transition path theory that a localized transition tube exists in one coordinate frame, but may not in another one (14). In our study, we used scaled coordinates to measure the relative changes of individual cells during EMT, which partially remove cell heterogeneity (e.g., cells with different initial cell size). In other contexts such as signal transduction studies people also use the fold-change scheme to remove cell heterogeneity (i.e., variation of basal expression levels of the signal transduction molecules).

c) Existence of slow degrees of freedom may complicate the dynamics along the projected coordinate to be non-Markovian. We performed Chapman-Kolmogorov test, which indicates that the dynamics along the 1D reaction coordinates is largely Markovian. In the revised Discussions, we suggest that more systematic studies would be needed, especially expanding the state space dimensionality will shed light on the coupling between the transition process and other cellular processes. It will be an exciting future direction. Notice that it is practically straightforward to record multi-dimensional single cell trajectories from the live cell imaging platform presented here, which would be generally very challenging or not feasible for a molecular system. We suggest that the quantitative imaging/analysis approaches and the experimental data will inspire further theoretical development of transition rate theories.

As a side note, Neupane et al.,(15) showed that “Protein folding trajectories can be described quantitatively by one-dimensional diffusion over measured energy landscapes”.

5. The authors demonstrated the two parallel pathways in EMT with the same starting and ending states (e.g., Figure 3e). While the reaction coordinates of transition state along one pathway (vimentin PC1 and morphology PC1) are intermediate between the E and the M states (the right panel of Figure 3e), those along the other pathway are not (the left panel of Figure 3e). What is the physical nature of this largely non-monotonic change? And if possible, what is the functional role? In perspective, cell operates in the multi-dimensional parameter space. What the authors have characterized is only the subset. Possibly, there exist additional but essential parameters that remain to be explored. This way, the non-monotonic change in the reaction coordinates may reflect the projection from a higher dimensional space onto the two-dimensional parameter plane. For instance, cell mechanics may be another set of key parameters that underlie EMT, which has been demonstrated to display non-monotonic changes during EMT (see Margaron et al., Biophysical properties of intermediate states of EMT outperform both epithelial and mesenchymal states (bioRxiv, 2019)). I'd suggest the authors discuss the finding in a broader context.

Thanks. The physical nature of this change might be that in some of the cells, the variation of morphology probably relies on the variation of vimentin. It is reported that vimentin can regulate the morphology through several ways. Vimentin is required for the mediation of Slug and Axl (16-19), and it can induce variation of cell morphology, motility and adhesion (19). Vimentin fibers regulate cytoskeleton architecture (18).

Though we can represent single cell trajectories in a very high dimensional composite feature space, this might be still insufficient to resolve the state of a cell. As you suggested, the two paths are probably the projection of higher dimensional space. We are going to explore it by including more features. And we add some discussion on this topic in the manuscript.

References

1. Zi Z, Chapnick DA, and Liu X (2012) Dynamics of TGF-β/Smad signaling. FEBS Lett 586(14):1921-1928.

2. Zhang J, et al. (2018) Pathway crosstalk enables cells to interpret TGF-β duration. npj Systems Biology and Applications 4(1):18.

3. Vizán P, et al. (2013) Controlling long-term signaling: receptor dynamics determine attenuation and refractory behavior of the TGF-β pathway. Science signaling 6(305):ra106.

4. Pincus Z & Theriot J (2007) Comparison of quantitative methods for cell‐shape analysis. Journal of microscopy 227(2):140-156.

5. Wang J, et al. (2008) Cellular phenotype recognition for high-content RNA interference genome-wide screening. Journal of Biomolecular Screening 13(1):29-39.

6. Yang J, et al. (2020) Guidelines and definitions for research on epithelial–mesenchymal transition. Nat Rev Mol Cell Biol 21:341-352.

7. Chang HH, Hemberg M, Barahona M, Ingber DE, and Huang S (2008) Transcriptome-wide noise controls lineage choice in mammalian progenitor cells. Nature 453(7194):544-547.

8. Gupta PB, et al. (2011) Stochastic state transitions give rise to phenotypic equilibrium in populations of cancer cells. Cell 146(4):633-644.

9. Wang W, et al. (2014) Dynamics between Cancer Cell Subpopulations Reveals a Model Coordinating with Both Hierarchical and Stochastic Concepts. PLoS ONE 9(1):e84654.

10. Mojtahedi M, et al. (2016) Cell Fate Decision as High-Dimensional Critical State Transition. PLOS Biology 14(12):e2000640.

11. Scheffer M, et al. (2009) Early-warning signals for critical transitions. Nature 461(7260):53-59.

12. Qiu X, et al. (2022) Mapping Transcriptomic Vector Fields of Single Cells. Cell (in press).

13. Zwanzig R (1992) Dynamic Disorder - Passage through a Fluctuating Bottleneck. J. Chem. Phys. 97(5):3587-3589.

14. E W & Vanden-Eijnden E (2010) Transition-path theory and path-finding algorithms for the study of rare events. Annu Rev Phys Chem 61:391-420.

15. Neupane K, Manuel AP, and Woodside MT (2016) Protein folding trajectories can be described quantitatively by one-dimensional diffusion over measured energy landscapes. Nature Physics 12(7):700-703.

16. Vuoriluoto K, et al. (2011) Vimentin regulates EMT induction by Slug and oncogenic H-Ras and migration by governing Axl expression in breast cancer. Oncogene 30(12):1436.

17. Ivaska J (2011) Vimentin: Central hub in EMT induction? Small GTPases 2(1):1436-1448.

18. Liu C-Y, Lin H-H, Tang M-J, and Wang Y-K (2015) Vimentin contributes to epithelial-mesenchymal transition cancer cell mechanics by mediating cytoskeletal organization and focal adhesion maturation. Oncotarget 6(18):15966.

19. Mendez MG, Kojima S-I, and Goldman RD (2010) Vimentin induces changes in cell shape, motility, and adhesion during the epithelial to mesenchymal transition. The FASEB Journal 24(6):1838-1851.